# Representing soil landscapes from digital soil mapping products – helping the map to speak for itself

David G. Rossiter<sup>1,2</sup> and Laura Poggio<sup>1</sup>

**Correspondence:** David G. Rossiter (david.rossiter@isric.org)

<sup>&</sup>lt;sup>1</sup>ISRIC-World Soil Information, Wageningen (NL)

<sup>&</sup>lt;sup>2</sup>Section of Soil & Crop Sciences, College of Agriculture & Life Sciences, Cornell University, Ithaca NY 14850 (USA)

**Abstract.** Since the earliest days of soil geography, it has been clear that soils occur in more-or-less clearly mappable bodies, within which soil forming factors have been either fairly homogeneous or in a regular pattern within the body, and between which there is usually a clear transition in one or more factors. This has been the basis for polygon-based soil mapping; make a concept map from landscape elements leading to a mental model of the landscape, confirm or modify it with strategically placed observations, find the transitions, delineate the soil bodies, and characterise them. By contrast, common methods of Digital Soil Mapping (DSM) predict per pixel over a regular grid, from training observations at pedon support. Accuracy assessment of DSM products has been at this "point" support, ignoring the existence of spatial soil bodies and the relations between pixels. Different approaches to DSM – datasets, model forms, analyst choices – result in maps with distinctly different patterns of predicted soil properties or types. Techniques from landscape ecology have been used to characterize spatial patterns of DSM products. The question remains as to how well these products reproduce the actual soil patterns at a given cartographic scale and categorical level of detail. Our approach is to help DSM maps to "speak for themselves" and thereby reveal spatial patterns that have been found by the DSM. We do this by grouping predictions at the individual pixel level, either (1) by aggregation based on property homogeneity using the supercells algorithm, or (2) by segmentation based on within-block property pattern similarity, using the GeoPAT suite of computer programs. Segments can be hierarchically clustered into groups of presumed soil landscape elements. Supercells and segments can be compared to existing soil maps, other land resource maps, and expert judgement. To the extent that the presumed soilscape patterns are reproduced, this is evidence that DSM has identified the soil landscape at the chosen scale. Since map users perceive patterns, and most land use decisions are for areas rather than pixels, we propose that DSM products be evaluated by their patterns as revealed by aggregation and segmentation, as well as by pointwise evaluation statistics.

## 20 1 Introduction

50

Digital Soil Mapping (DSM) is a general term for the creation of digital maps of soil classes or properties by fitting geostatistical (Webster and Oliver, 2008), statistical learning (Hastie et al., 2009), or similarity-based (Zhu and Turner, 2022) models between observations of soil classes or properties at known locations and a set of environmental covariates representing soil-forming factors. This term has also been applied to soil maps based on GIS overlay of presumed soil-forming factors, for example, the eSOTER approach (Dobos et al., 2019). Some authors follow the review of Scull et al. (2003) and refer to this as Predictive Soil Mapping (PSM), although since all soil mapping is by nature predictive, this seems to be a less specific term. Since its formal introduction by McBratney et al. (2003) DSM has been applied worldwide at a wide range of scales and target classes and properties; see reviews by Mulder et al. (2023), Arrouays et al. (2020) and Nenkam et al. (2024) and future perspectives by Lagacherie (2025). DSM is a semi-automated digital form of landscape analysis as used in traditional soil survey to identify distinct soils from environmental covariates (Hole and Campbell, 1985; Hudson, 1992). However, as DSM predicts at the pixel level, it ignores spatial relations. As Vaysse and Lagacherie (2017) aptly state, "DSM products are simplified representations of more complex and partially unknown patterns of soil variations", where this "simplification" is reducing landscapes to individual pixels.

DSM products are routinely and (almost) exclusively evaluated by point-based evaluation statistics, including the cross-validation mean error (ME), root-mean squared error (RMSE), proportion of variance explained (1:1 R²) and the model efficient coefficient (MEC) (Helfenstein et al., 2024, Formulas 2–4). These are almost never based on probability or even representative training (i.e., cross-validation) observations (Piikki et al., 2021). Point-based evaluation ignores the existence of soil bodies that form a pattern over the landscape. Maps with distinctly different patterns of predicted soil properties or types can result from different approaches to DSM, see for example Rossiter et al. (2022) and Poggio et al. (2010a). We propose to also evaluate DSM products by their patterns, as revealed by aggregation and segmentation of the gridded maps into areas with more or less homogeneous internal composition of soil properties.

Soil geographers conceive of the soilscape as a continuum in 3D, with the vertical dimension (soil profile) defining a *pedon* (Soil Survey Staff, 1999, p. 11). The pedon has a horizontal dimension sufficient to show the local variability of horizons and properties, e.g., cyclic or irregular horizons. Pedons are connected laterally into relatively homogeneous *polypedons* (Johnson, 1963), within which the soil-forming factors and hence the pedons are within some defined limits. The transition zones between polypedons are marked as borders between natural soil bodies according to those limits, which may be abrupt or smooth (Lagacherie et al., 1996), according to the spatial pattern of the soil-forming factors. Figure 1 shows a typical conceptual model from a detailed Order 2 soil survey in the USA, design scale 1:12 000 (minimum mappable area 0.576 ha). The transitions between polypedons in this scene are due to parent material, topography, and hydrology.

The pattern of the distribution of polypedons on the landscape make up the *soilscape*. The classic example is the catena as defined by Milne (Milne, 1935) as: "a sequence of distinct but pedogenetically-related soils that are consistently located on specific slope facets, giving recurrent topographically-associated soil pattern" (Borden et al., 2020), We would hope that a DSM-produced map of a catena would clearly show these elements and their transitions.

Figure 1. Conceptual block diagram, Otsego County NY (USA)

Source: https://www.nrcs.usda.gov/publications/NY-2010-09-28-14.png

In traditional expert-based soil class mapping (Hudson, 1992) the landscape is segmented according to the mapper's conceptual model of soil-landscape relations, and by examination of external clues, notably relief, vegetation, and land use, and by augering or full profile examination. DSM replaces the conceptual model with correlative relations with digital coverages meant to represent, at least in part, one or more of the seven "SCORPAN" predictive factors of McBratney et al. (2003). In this widely-cited paper they briefly describe as these factors as: s: soil, other properties of the soil at a point; c: climate, climatic properties of the environment at a point; o: organisms, vegetation or fauna or human activity; r: topography, landscape attributes; p: parent material, lithology; a: age, the time factor; n: space, spatial position. The time factor accounts for the changing climate, organisms (including human activities) and relief over the time of soil formation. In practice, the time factor has proven quite difficult to represent by digital coverages. Note that these are correlative, not necessarily causative, and are used to build a predictive model for mapping, not (at first) to understand pedogenesis. Thus in DSM there is no longer an explicit relation with the soil landscape, but it is hoped that the implicit correlative relations, based on representative covariates, can find these.

The concept of areas with distinct patterns of contrasting soils goes back to the "soilscape fabrics" from the soilscape analysis of Hole (1978) and the "soil combinations" of Fridland (1974). With increasingly-detailed cartographic scales and categorical definitions of soil types increasingly finer patterns can be shown. Conversely, at coarser scales and broader categories patterns

are necessarily more general. As Fridland puts it, "Soil combinations consist of elementary soil areas which are genetically linked to various degrees and which produce a definite pattern in the soil mantle... Multiple spatial repetition of a certain soil combination or several soil combinations alternating in a definite order creates various forms of structures of the soil mantle." An example of a fine-scale soil pattern is the pit and mound topography found on a hillslope in southwest Poland by Pawlik et al. (2024).

In traditional soil mapping, these areas with sufficiently homogeneous soils or patterns of them at a given cartographic scale are the units that are delineated on the map. However, as Fridland explains: "The structure of the soil mantle and soil combinations are in their essence not cartographic but genetic-geographic concepts, even though they constitute a basis for elaborating cartographic units." This implies that the resulting soil properties distributed vertically in the profile, as products of pedogenesis, can be the basis for map units. Therefore, if at each pixel DSM accurately predicts a sufficiently rich set of properties over the soil profile, these should be grouped on the DSM map as recognizable cartographic units.

Within a mappable soilscape segment, there will of course be variability, ranging from some smaller deviations from a central concept (typical soilscape position and pedon), to a mixture of contrasting pedons, in National Resource Conservation Service (NRCS) soil survey terms a *complex*. Since predictions made by DSM are per pixel, it may be possible to resolve these complexes into their components at the pixel scale, if that is fine enough to match the pattern within the complex. If this is the case, our evaluation of the DSM product should identify this.

Digital Soil Mapping (DSM) products show predicted values of soil properties or classes at each pixel of a regular, more or less fine grid, either as the centre point or a block average of the area covered by the pixel. DSM typically predicts multiple soil properties at a set of standard depth slices. Although some DSM methods use covariates in areas around a pixel, they do not enforce any relation between adjacent pixels. These relations are particularly important in soil hydrology models. Thus, the question is to what degree the pixels of DSM products at various resolutions can be aggregated into groups to realistically represent a soil landscape, whether the soilscape segment is relatively homogeneous in its properties or represents an association or complex. Intuitively, if the soil-forming factors responsible for a polypedon are also spatially associated in the covariates used in DSM, the relations between pixels should occur as a by-product of per-pixel DSM. More abrupt transitions in the covariates should be reflected in the predictions. The pattern of the pixels should therefore represent the soil landscape. The question is, does the DSM product show these relations?

In this study, we examine two methods to assess the success of DSM in reproducing a soil landscape. The first method is to aggregate the individual predictions from pixels into more or less homogeneous contiguous groups of pixels referred to *supercells*, following methods used in image processing, where these are called *superpixels* (Nowosad and Stepinski, 2022). This can be based on single properties and depth layers, or, more usefully, on the multivariate collection of DSM-predicted properties at a pixel. We explain the aggregation algorithm in §2.1.

The second method is applied at coarser scales, where the homogeneity of properties within some larger area may not be possible or even desirable. This has led to the concept of landscape *segments*, defined by the co-occurrence pattern, referred to as a *signature*, of a group of contrasting pixels of a class map, within a predefined size of the segment. Segmentation was developed by geographers to find similar land cover patterns for ecoregionalization (Nowosad and Stepinski, 2018). In that

case, the pixels represent land cover classes. The aim is not homogeneity of land cover, rather, homogeneity of the land cover pattern within some analyst-defined area. The relation to a soil cover pattern is obvious, and corresponds well to concepts such as the catena or soil associations.

These two concepts, aggregation and segmentation, can be related to traditional soil survey practice. Depending on the scale of the analysis (for DSM, the horizontal resolution, for traditional soil survey the minimum delineation size) and the inherent scale of the soil landscape, we may expect to see homogeneity at the level of map delineations containing dominantly one soil type within defined limits at a detailed categorical level (e.g., soil series, the lowest level of Soil Taxonomy); this is called a *consociation* in the US soil survey (Soil Science Division Staff, 2017). This is where aggregation is useful, to identify homogeneous components that can be mapped as separate units. At a coarser scale we may expect a regular pattern of contrasting soil types forming a soil *association*, or a fine-scale pattern of contrasting soils forming a soil *complex*. This is where segmentation is useful, to form mapping units with consistent heterogeneous composition, These terms from the US soil survey are well-explained, with examples, by Van Wambeke and Forbes (1986).

Segmentation requires that DSM maps of continuous predictions be classified, i.e., sliced according to analyst-defined class limits. The classes can correspond to meaningful classes for soil management, or can be based on laboratory precision. They can be wider (more general) or narrower, roughly corresponding to cartographic detail. Clearly, the classification can greatly influence segmentation. This is also the case when segmenting land cover classes. We explain the segmentation algorithm in §2.2.

Once a segmentation has been performed, the segments can be clustered according to their similarity of internal pattern, i.e., the signature of the segment. These can then be examined to find similar soil landscape elements in different parts of the map. We explain the clustering procedure in §2.3.

The objective of this study is present methods to create possible soil landscape units from DSM products, by both aggregation and segmentation, and then to cluster the segments to identify similar soil-landscape units within the map. These proposed units can be characterized statistically by their composition, internal variability and differentiation from their neighbours, as well as evaluated visually. We first describe the methods (§2) and then apply them to three case studies (§3 BIS-4D Netherlands, §4 SoilGrids v2.0 global, §5 SOLUS 100 m USA) corresponding to different DSM projects at various resolutions and extents. Finally, we discuss (§6) how these methods can be used in the evaluation of DSM products.

#### 130 2 Methods

115

120

We contrast two approaches to helping the map to "speak for itself": aggregation based on homogeneity of properties (§2.1), and segmentation based on patterns of classified properties within segments (§2.2).

## 2.1 Aggregation

Aggregation seeks to find contiguous groups of pixels with relatively homogeneous property values, either single or multivariate. This is implemented by the supercells R package (Nowosad, 2025), which uses the Simple Linear Iterative Clustering

(SLIC) image-processing algorithm (Nowosad and Stepinski, 2022), with the improvement that an appropriate data distance measure and function for cluster averaging can be defined by the analyst. For multivariate aggregation there must be a distance measure defined in multivariate space. A common choice, used here, is the Jensen-Shannon divergence (Lin, 1991), which quantifies the distance between two histograms by the deviation between the Shannon entropy of the combination of two unior multivariate histograms and the mean of their individual entropies.

The supercells function is controlled by several parameters that have a large effect on the results. First and most important is *compactness*, which trades off internal homogeneity of the supercells with their geometric compactness. The absolute compactness value depends on the range of input pixel values and the selected distance measure. A large value prioritizes spatial distances between pixels and superpixel centres (more geometric compactness), whereas a smaller value prioritizes distances in feature space (more property homogeneity). Second is the approximate number of supercells, *k*. This should correspond to the number of landscape segments expected in the study area, at the design scale of the corresponding polygon map. Third is the minimum supercell size, *minarea*. This should correspond to a minimum mappable area or a minimum size needed for an application, e.g., land management or stratified sampling.

The quality of the aggregation can be evaluated by the standard deviation or coefficient of variability of each property in the supercell. As supercells decrease in size, these measures will necessarily have smaller values.

## 2.2 Segmentation

155

Segmentation seeks to find contiguous groups of blocks of grid cells with similar internal patterns of pixels, which represent soil classes or properties, either univariate or multivariate. The GeoPAT implementation of segmentation compares patterns within square blocks of at least 10 x 10 pixels and then joins adjacent blocks with similar internal patterns into rectilinear segments. Larger blocks can be specified by the analyst, according to the desired scale of the analysis.

Segmentation proceeds as follows. The first step is to select classified soil properties and their depth slices to represent soil individuals at each pixel. The second step is to find the co-occurrence pattern of the pixels within pre-defined grid cells. The third step is to aggregate grid cells with similar internal spatial patterns into larger units, sufficiently distinct from neighbouring units in terms of their internal spatial patterns. Finally, the result is evaluated by its segmentation statistics, namely, inhomogeneity within the segment and isolation of the segment from its neighbours. The segmentation can be inspected by expert judgement, perhaps comparing with conventional soil maps, to evaluate how well it represents the soil landscape at the selected cartographic scale.

For segmentation, we use the GeoPAT suite of stand-alone Unix programs (Jasiewicz et al., 2015). These are invoked in sequence, via the R system function, to obtain a segmentation and an evaluation of its quality. GeoPAT has been used successfully to segment categorical rasters such as land cover maps (Jasiewicz et al., 2018) and for global ecoregionalization based on multiple environmental factors (Nowosad and Stepinski, 2018). Figure (2) shows the segmentation workflow using GeoPAT.

Several parameters control the signature computation of the gpat\_gridhis "create a binary grid of signatures" program. Two related parameters are size and motifel. The first is the size of the output grid cell of the segmented map. This must

Figure 3.13: Workflow path for segmentation

**Figure 2.** GeoPAT segmentation workflow. gpat\_gridhis: "create a binary grid of signatures'; gpat\_segment "segment a grid-of-scenes"; gpat\_segquality "compute quality metrics of a segmentation"; gpat\_gridts not used. Source: (Netzel et al., 2018)

be at least 10 x 10 pixels of the source DSM. Thus, the segmentation is of similar patterns within an output grid cell and its neighbours. This dictates the largest equivalent map scale at which soilscape patterns (groups of output grid cells) can be discerned. The second is the "Motif Element", referred to as the *motifel*, defined as the size of the window within which the pattern will be computed. This must be at least as large as the size, but could be larger to account for edge effects in the pattern. Two important threshold parameters for joining grid cells into segments are lthreshold to control the sizes of segments and uthreshold to prevent the growth of inhomogeneous segments.

Another important option for <code>gpat\_gridhis</code> is the signature type within each grid cell, default <code>cooc</code>, "spatial cooccurrence of categories". This characterizes signatures with a "colour" co-occurrence histogram, a variant of the Gray-Level Co-occurrence Matrix (GLCM) used to characterise texture in greyscale images (Haralick et al., 1973; Hall-Beyer, 2017). In GeoPAT, discrete greyscale numbers, as in GLCM, are replaced by cell classes. A separation of one pixel is used to calculate the co-occurrence histogram, which then represents the spatial pattern within a grid cell. Related to this is the normalization type, default <code>pdf</code> "probability distribution function", which is recommended for the <code>cooc</code> signature type. This harmonizes the signatures from different motifels.


Grid creation requires the selection of grid sizes. To evaluate DSM products we select these based on their correspondence to nominal map scales, using the Vink definition of a minimum legible delineation (MLD), i.e., the smallest area that can

be displayed on a printed map, of  $0.25~\rm cm^2$  at map scale, i.e., a grid cell side of  $0.5~\rm cm$  (Vink, 1963). The Optimal Legible Delineation (OLD) is conventionally defined as 4 x MLD (Forbes et al., 1982). This is a delineation size which is easily legible and still small enough to be relatively homogeneous. In conventional mapping the map scale should be set so that the soil pattern is on average able to be shown by OLD-sized polygons. In segmenting DSM products we hope that most segments are at least as large as the OLD.

To determine the Minimum Legible Area (MLA) and corresponding side on the ground, the MLD is multiplied by the scale number (denominator of the scale ratio). For example, at 1:200 000 the MLA is 100 ha, with a side of 1 km. Signature computation requires at least 100 pixels from the DSM map in order to produce a reliable signature, i.e., the minimum edge of the segmentation grid (the "shift" parameter) must be 10 times the original DSM resolution. For example, a 25 x 25 m DSM product can only be segmented at 250 x 250 m or coarser (6.25 ha), corresponding to the MLA of a 1:50 000 scale map. To match a 1:200 000 map (MLA 100 ha), the 25 x 25 m pixel must be aggregated 40 times per side, i.e., 1 km x 1 km. These concepts are comparable to concept of soil survey orders in the USA soil survey (Soil Science Division Staff, 2017, Chapter 4) and the "resolutions and extents for DSM" of (McBratney et al., 2003, Table 1).

The segmentation phase in GeoPAT is implemented by the <code>gpat\_segment</code> "segment a grid-of-scenes" program. This groups grid cells based on their motifel signatures computed by <code>gpat\_gridhis</code>. Segments have a "brick" topology, in which square grid cells are arranged in alternating layers with each layer is shifted by one-half the size of the motifel. Thus, the analysed area (i.e., the MLA) is four times the motifel size.

Segment homogeneity is characterised by their normalised Shannon entropy H, defined as:




$$H = -\sum_{i=1}^{n_y} p_i \log_{n_z} p_i \tag{1}$$

where  $p_i$  is the proportion of the segment in class i,  $n_z$  is the number of possible classes, and these are summed over all  $n_y$  pixels in the grid cell. Using the logarithm to base  $n_z$  normalizes the entropy to the unit range regardless of the number of possible classes, so that 0 indicates complete homogeneity, i.e., one class for the entire segment. By contrast, 1 indicates maximum heterogeneity, i.e., all classes are equally represented in the segment. This only depends on class composition, not on pattern, even though the latter is the basis for segmentation.

Segmentation quality is measured with the gpat\_segquality "compute quality metrics of a segmentation" program. This produces two quality measures: (1) the inhomogeneity within each segment, and (2) the isolation of each segment from its neighbours. Inhomogeneity measures the degree of mutual dissimilarity between a segment's motifels, on a [0...1] scale, where smaller values correspond to more homogeneous and less internally diverse segments. Isolation is the average dissimilarity between a segment and its immediate neighbours, on a [0...1] scale, where larger values correspond to segments that are more isolated from their neighbours. These measures depend on the pattern, not just the class composition, of segments. The most successful segmentation would have the smallest inhomogeneity and largest isolation.

Figure 3.27: Workflow path for a clustering of segments (regions)

**Figure 3.** GeoPAT clustering workflow. gpat\_polygons "calculate numerical signatures of irregular regions"; gpat\_distmtx "compute a distance matrix between a collection of scenes". Source: (Netzel et al., 2018)

## 2.3 Clustering



Once segments are created, their internal patterns can be characterised by the same signature methods used to perform the segmentation. Figure (3) shows the workflow for clustering in GeoPAT. The <code>gpat\_polygons</code> "calculate numerical signatures of irregular regions" program computes the signature within each segment. The distance between these signatures is then computed by the <code>gpat\_distmtx</code> "compute a distance matrix between a collection of scenes" program. Here we used the default Jensen-Shannon divergence, because it is easily interpretable on a [0...1] scale and is not sensitive to extreme values (Lin, 1991). The segments can then be clustered on the basis of their distance measures by many clustering algorithms; see the comprehensive description in Gan et al. (2021). Here we use hierarchical clustering, as implemented by the R function <code>hclust</code> using Ward's linkage with squared distances to produce a dendrogram. This is cut at an analyst-determined number of classes to represent groups of internal homogeneity of segments. We chose Ward's with squared distances (Ward's D2) to minimize within-cluster variance. This minimizes the loss of information associated with each merging as the dendrogram is built bottom-up. There other choices in both the distance measurement and clustering linkage method, here we want to illustrate the clustering concept, not compare clustering methods.

## 3 Case Study 1 – BIS-4D (Netherlands)

BIS-4D ("Bodeninformatiesysteem 4-Dimensional") (Helfenstein et al., 2024) is a high-resolution (25 m horizontal, six depth slices vertical) soil modelling and mapping platform for the Netherlands. The 3D are geographic space and depth along the

soil profile. The fourth dimension is time, applied only to soil organic matter (SOM), which we ignore here by using only the most recent SOM map. Predicted properties are clay, silt, sand and SOM concentrations %, bulk density  $g \, cm^{-3}$ , pH in KCl, total N mg kg<sup>-1</sup>, oxalate-extractable P mmol kg<sup>-1</sup>, and cation exchange capacity mmol(c) kg<sup>-1</sup>. Depth slices are the *GlobalSoilMap* standard 0–5, 5–15, 15–30, 30–60, 60–100 and 100-200 cm (Science Committee, 2015). Each map is accompanied by uncertainties (quantiles and 90% prediction interval). We did not use these in this analysis, only the mean predictions. Coverages in the *GeoTIFF* format are free to download and use, and can be directly read into the terra R package (Hijmans et al., 2025).

BIS-4D is fairly accurate at point support, as assessed by cross-validation (Helfenstein et al., 2024, Tables 7, 8), due to a very dense sampling network and the country-specific covariates used in the DSM. For example, the 10-fold cross-validation average for all predictions of pH had a median ME of -0.023 pH, median RMSE of 0.72 pH, and a median MEC of 0.72. For clay these accuracy statistics are 0.42%, 7.7%, and 0.78, respectively. Visual inspection of layers agrees well with traditional 1:50 000 scale polygon soil maps (Steur and Heijink, 1980; Brouwer et al., 2021) and expert views of the soil landscape.

We selected a 40 x 40 km test area (Figure 4), because of its diverse soil-forming environments, including river clays of various ages and compositions, sandy push moraines, organic soils in glacial depressions, and coversands.

## 3.1 Aggregation





The supercells algorithm can work directly on raster stacks of the terra package. All 54 maps (nine properties, each with six depth layers) were combined in a SpatRaster raster stack. Since the values and ranges are not compatible, the Jensen-Shannon divergence was used to evaluate the distance in feature space between pixels and supercell centres. In this landscape there are non-compact (extended) features parallel to the river, in the fen areas and along the push moraines, so after some experimentation a low *compactness* value (0.2) was selected. We selected a minimum mappable area of 10 ha, equivalent to the 1:50 000 design scale of the Dutch conventional soil map, using the Cornell definition of 0.4 cm<sup>2</sup> minimum legible area on the map (Forbes et al., 1982). Thus the *minarea* parameter was set to 1,600 pixels, each of 25 m x 25 m.

Figure 5 shows the supercells (outlined in black) with several properties as a background. Note that the supercells in all maps are the same, but the mean values of each property within the supercells are different. The median size of the 270 supercells was 433 ha, ranging from 104 to 5 044 ha, with a strongly right-skewed distribution. Aggregation clearly shows the differences between soil bodies, with some properties being more prominent in certain supercells.

To evaluate the quality of the aggregation, we computed the standard deviation of each property within each supercell (Figure 6). These are quite low for clay and SOM, and for pH with some small areas with notable exceptions. Bulk density is less successfully aggregated. The high standard deviations in a supercell occur when that property has a small contribution to the computation of Jensen-Shannon divergence in that supercell.

## 3.2 Segmentation

Since gpat\_gridhis requires class maps, to illustrate this method we classified the soil property maps as follows: bulk density by  $0.1~{\rm g~cm^{-3}}$ , CEC by  $25~{\rm mmol(c)~kg^{-1}}$ , clay, silt, sand concentrations by 5%,  $P_{\rm ox}$  by  $4~{\rm mmol~kg^{-1}}$ , pH by  $0.1~{\rm g~cm^{-3}}$ 

Figure 4. Semi-detailed soil map of the Netherlands, design scale 1:50 000 (part).

Source and detailed legend: Ministerie van Volkshuisvesting en Ruimtelijke Ordening (2024).

General legend: Dark and medium green: river clays with different clay concentrations; Light green: glacial depression sediments; Brown, pink: push moraines with varying sand and gravel sizes; Yellow: wind-blown sands; Purple: peat.

units, SOM concentration by 4%, and total N by 1000 mg kg<sup>-1</sup>. In practice, the map evaluator would select class limits to correspond to the desired precision and thresholds for interpretations or models. The class widths can not be finer than the precision of the corresponding laboratory analyses, which usually are more precise than the precision needed for applications.. For example, the guidelines for liming in New York State (Ketterings and Workman, 2023) recommend based on a precision of 0.1 pH, although the recommended laboratory method has a precision of 0.01 pH. Another consideration is the precision of the DSM. In this example pH was predicted with an overall RMSE of 0.72 pH, so perhaps the classes should have been defined more coarsely than the selected 0.1 pH.

The minimum grid size for segmentation (10 x 10 pixels) is 250 x 250 m (62.5 ha), corresponding to a 1:158 000 scale map by the Vink definition, or 1:125 000 by the Cornell definition, as explained in §2.2. Segmentation at this resolution is expected

Figure 5. Results for selected properties of aggregation by supercells algorithm using all properties and layers

to more closely match the 1:200 000 generalised soil map of the Netherlands (Haans, 1965) than the 1:50 000 semi-detailed map shown in Figure 3.

## 3.2.1 Univariate segmentation of individual maps


To examine the effect of grid size, we segmented all properties at all depths, individually, at the minimum possible grid cell size, i.e.,  $10 \times 10$  and at several multiples corresponding to nominal map scales 1:100 000, 1:200 000, 1:400 000, and 1:800 000, respectively. The next coarser resolution (1:1'600 000) resulted in only one or two segments and so was not used in this test area, only for the entire Netherlands (§3.2.4, below). Table 1 shows the results for one run of the segmentation process. Note that because of the random aspects in the algorithm other runs give slightly different results. Comparing the finest segmentation to the single grid cell at resolution,  $0.625 \text{ km}^2$ , we see that many segments were of one or two grid cells. This pattern was mostly very fine, with a few large segments for most single properties. Each quadrupling of the grid area resulted in larger segments, but these were not simply groupings of the previous segments. In general, the various depth slices

Figure 6. Standard deviations for selected properties of aggregation by supercells algorithm using all properties and layers

of pH were the least successfully grouped into larger segments, whereas Pox and SOM were able to form large segments. The clay and silt particle-size classes, CEC, and bulk density were intermediate. This may be in part to the selected class limits for the properties, as well as intrinsic spatial variability.

# 3.2.2 Multivariate segmentation of individual properties, all depth slices



We then performed a multivariate segmentation using all depth slices of single properties. By default, GeoPAT normalizes each layer and by default weights them equally. In this mode, a motifel must meet the threshold conditions for all input layers to be joined to a segment. In this way the segmentation is meaningful for the whole profile. Because of the different spatial structures of the properties at each depth slice, it was expected that the segmentation would be finer at each scale than for individual depth slices, i.e., it would be more difficult to merge grid cells. The results for one run of the segmentation process are shown in Table 2. Contrary to our expectations, the median number of segments were all smaller than those for the corresponding property's single depth slice segmentations. This shows that using the multivariate measure of similarity with the same model parameters

| Cell size      | nominal scale | maximum segments property | $n_{max}$ | minimum segments property | $n_{min}$ | $n_{median}$ | median area km² |
|----------------|---------------|---------------------------|-----------|---------------------------|-----------|--------------|-----------------|
| $10 \times 10$ | 1:100 000     | pH 05-15 cm               | 3 867     | SOM 15-30 cm              | 534       | 2 366        | 0.68            |
| $20\times20$   | 1:200 000     | pH 05–15 cm               | 832       | Pox 60–100 cm             | 138       | 621          | 2.58            |
| $40 \times 40$ | 1:400 000     | sand 05-15 cm             | 205       | SOM 15-30 cm              | 30        | 153          | 10.46           |
| $80 \times 80$ | 1:800 000     | sand 60 – 100 cm          | 52        | SOM 30-60 cm              | 8         | 40           | 40.00           |

Table 1. Results of univariate segmentation: each property at each depth slice separately (54 segmentations).  $n_{max}$ : maximum number of segments found by all properties;  $n_{min}$ : number of segments found for the "minimum segments" property;  $n_{median}$ : median number of segments found by all properties.

| Cell size      | nominal scale | maximum segments property | $n_{max}$ | minimum segments property | $n_{min}$ | $n_{median}$ | average area km <sup>2</sup> |
|----------------|---------------|---------------------------|-----------|---------------------------|-----------|--------------|------------------------------|
| 10 × 10        | 1:100 000     | pН                        | 3 114     | SOM                       | 162       | 1 633        | 0.98                         |
| $20 \times 20$ | 1:200 000     | pH                        | 668       | SOM                       | 38        | 403          | 3.97                         |
| $40 \times 40$ | 1:400 000     | sand                      | 174       | SOM                       | 12        | 111          | 14.41                        |
| $80 \times 80$ | 1:800 000     | sand                      | 44        | SOM                       | 5         | 37           | 43.24                        |

Table 2. Results of multivariate segmentation: all depth slices for each property (9 segmentations).  $n_{max}$ : maximum number of segments found by all properties;  $n_{min}$ : number of segments found for the "minimum segments" property;  $n_{median}$ : median number of segments found by all properties.

allows for larger areas with the same internal pattern. Again, the maps of pH and sand could only be grouped into small segments, and SOM into the largest segments.

Figure 7 shows the segmentation based on whole-profile bulk density at the finest scale (nominal 1:100 000), overlaid on the six depth slices. There is a clear landscape pattern. The sandy areas with higher bulk density, as well as the medium bulk densities in the older river clays, are mostly collected into large polygons. The fine details in peat areas and younger river sediments are also captured.




Figure 8 compares the segment boundaries for the multivariate segmentation by bulk density over the whole profile, at four resolutions overlaid on the Dutch 1:50 000 soil survey polygons. It is clear that the necessarily larger polygons resulting from the coarser segmentations miss important differences, and that the 1:100 000 segmentation finds quite small areas, mostly just one grid cell, within soil bodies. The 1:200 000 segmentation (i.e., shift size 20, i.e., 0.5 km² grid cells) matches well with many soil map boundaries. Note however that the Dutch soil survey map units are defined by many properties, not just bulk density.

Figure 9 shows the success of the segmentation based on bulk density over the whole profile at two design scales. This is evaluated by the internal inhomogeneity of each segment and the difference of this from its neighbours, i.e., the isolation. For example, at both scales the polygon at upper left, representing part of the sandy uplands (the Utrechtse Heuvelrug), has low inhomogeneity (similar internal composition of the bulk density profiles of its pixels), and high isolation, i.e., its internal

**Figure 7.** Segmentation based on bulk density over the whole profile (red lines), overlaid on soil map polygons (grey lines). Design scales left to right, top to bottom: 1:100 000, 1:200 000, 1:400 000, 1:800 000

average area km<sup>2</sup> Cell size nominal scale number of segments inhomogeneity isolation  $10 \times 10$ 1:100 000 985 0.105 0.277 1.62 277  $20 \times 20$ 1:200 000 0.085 0.250 5.78  $40 \times 40$ 1:400 000 77 0.081 0.199 20.78  $80 \times 80$ 1:800 000 22 0.086 0.184 72.73

Table 3. Results of segmentation based on the bulk density profile.

composition is quite different from that of its neighbours. This landscape segment has been well-identified at both scales, because it has such a distinctive bulk density profile (very high throughout) in contrast to its neighbours.

Table 3 shows that as the segmentation becomes coarser the inhomogeneity and isolation both decrease, i.e., segments are internally more consistent in their internal patterns and less isolated from their neighbours. This illustrates the effect of geographic generalisation.

# 3.2.3 Multivariate segmentation with selected properties and depth slices

Although BIS-4D predicts each property separately, the soil as a natural body is of course more than a stack of individual properties, and this is recognized by the concept of diagnostic horizons and properties in modern soil classification systems, and

**Figure 8.** Segmentation based on bulk density over the whole profile (red lines), overlaid on soil map polygons (grey lines). Design scales left to right, top to bottom: 1:100 000, 1:400 000, 1:800 000, 1:1'600 000

soil series in detailed conventional soil mapping. To see if segmentation of BIS-4D can identify these, we selected properties and depth slices to represent the profile. These were selected to match with expected diagnostic horizons and series differences in the test area. In other contexts the choices would be linked to the key soil properties and depth slices which differentiate the major soil types in that area. Using all 54 layers results in an impractical Jensen-Shannon divergence, hence we selected key properties at key depths: (1) pH, clay, silt, SOM at 0-5 cm; (2) clay and bulk density at 15-30 cm; (3) CEC at 30-60 cm; and (4) sand and SOM at 100-200 cm. The reason for including SOM of the deepest layer was to distinguish thick peats, and for sand of that same layer is to distinguish thick dune sands.

Table 4 shows the results for one run of the segmentation process. The segment counts at each scale are much smaller, and thus the segment areas are larger, than for individual properties and depth slices, and also for individual properties over the

**Figure 9.** Evaluation of segmentation based on bulk density over the whole profile at the 1:100 000 (top) and 1:400 000 (bottom) design scales. Note the different colour ramps for the two scales

whole profile, compare with Table 3. This follows the tendency observed for using full profiles of single properties, compared to single depth slices (§3.2.2).



Figure 10 shows the segment boundaries from this segmentation at the 1:400 000 design scale, overlaid on the properties and depth slices used to compute the segmentation. Many of the segments correspond to landscape features shown in the conventional soil map of Figure 4, although constrained to the rectilinear shape and minimum grid cell size. For example, segment 2 covers both the sandy push moraines, and segment 10 most of the lower Rhine floodplain. However, because the different properties and depths have different segmentations when considered independently, some obvious soil landscapes are not well-represented because the segmentation must consider all the properties and depths. For example, the areas with thick peat as shown on the 100-200 cm SOM map are not separated into segments, but rather included in larger segments. This suggests that the algorithm will have difficulty segmenting on the basis of multiple properties which are selected to represent major profile types.

| Cell size      | nominal scale | number of segments | inhomogeneity | isolation | average area km <sup>2</sup> |
|----------------|---------------|--------------------|---------------|-----------|------------------------------|
| $10 \times 10$ | 1:100 000     | 525                | 0.431         | 0.630     | 3.05                         |
| $20\times20$   | 1:200 000     | 118                | 0.366         | 0.574     | 13.56                        |
| $40 \times 40$ | 1:400 000     | 40                 | 0.365         | 0.566     | 40.00                        |
| $80 \times 80$ | 1:800 000     | 6                  | 0.414         | 0.569     | 266.67                       |

**Table 4.** Results of multivariate segmentation: with selected properties and depth slices

**Figure 10.** Segmentation based on selected properties and depth slices, overlaid on DSM of selected soil properties, 1:400 000 design scale. Legends not shown. Scale is from yellow (low values of the property) to dark blue (high values). Segments are labelled with their numbers.

# 340 3.2.4 Segmentation over a large area

To determine whether segmentation could be applied over a larger area than the 40 x 40 km test area, we segmented the BIS-4D product for the entire land area of the Netherlands ( $\approx 33~240~\mathrm{km}^2$ ) using all depth slices for three properties, at the three most

| Property           | $n, 40 \times 40$ | $n,80 \times 80$ | $n, 160 \times 160$ |
|--------------------|-------------------|------------------|---------------------|
| pН                 | 2240              | 601              | 161                 |
| Bulk density       | 1344              | 358              | 100                 |
| Clay concentration | 1444              | 462              | 143                 |

**Table 5.** Results of multivariate segmentation: all depth slices for selected properties, entire Netherlands. n = number of segments

**Figure 11.** Segmentation by whole-profile pH of the Netherlands at 1:800 000 (left) and 1:1'600 000 (right) nominal scales, overlaid on the pH 15–30 cm DSM product

general scales. The results are shown in Table 5. Interestingly, there is quite some difference in segment numbers among these properties. Bulk density (classified units of  $0.1 \ \mathrm{kg} \ \mathrm{m}^{-3}$ ) forms the fewest segments, whereas pH (classified in units of  $0.1 \ \mathrm{pH}$ ) forms the most segments. These results are partly due to the classification precision, as well as the spatial pattern of the properties.

Figure 11 shows the segmentation by pH (classified in units of 0.1 pH) of the entire Netherlands at the two most general scales. For this extent the coarsest segmentation seems most useful for understanding the generalized country-wide soil pattern. For example, the two push-moraine sand ridges (Utrechtse Heuvelrug and De Veluwe) are identified as one segment, as is most of the reclaimed marine clays of Flevoland. The complex pattern of low and medium pH in North Brabant is also identified as one generalized soil landscape.

## 3.2.5 Segmentation parameters



Segmentation is greatly affected by the two thresholds. Table 6 shows the results for one run of the segmentation process using all depth slices for clay concentration with the default lower and upper thresholds (0.1 and 0.3, respectively), compared with a

| Cell size      | nominal scale | n (conservative) | n (liberal) | reduction factor |
|----------------|---------------|------------------|-------------|------------------|
| $10 \times 10$ | 1:100 000     | 1 175            | 563         | 2.09             |
| $20\times20$   | 1:200 000     | 481              | 157         | 3.06             |
| $40\times40$   | 1:400 000     | 143              | 61          | 2.34             |
| $80 \times 80$ | 1:800 000     | 40               | 12          | 3.33             |

**Table 6.** Conservative and liberal segmentation, all depth slices of clay concentration.

more liberal (easier segmentation) thresholds (0.3 and 0.8, respectively), at several resolution. Using these liberal segmentation parameters reduces the number of segments between two- and three-fold. In effect, the more liberal segmentation at a finer scale is comparable to the more conservative one at the next-coarser scale.

This is illustrated in Figure 12, which shows the multivariate segmentation of the test area on the basis of clay concentration at all depth slices at nominal 1:100 000 and 1:200 000 scale with default and more liberal thresholds. The thresholds can be adjusted by the analyst to match known soil-landscape components. This is an example of "helping" the DSM product to "speak for itself".

## 3.3 Clustering


Hierarchical clustering was applied to the segments of Figure 10, i.e., based on properties and depth slices selected to represent the profiles of the major soil types. The resulting dendrogram is shown in Figure 13. Note the large separation in internal patterns between the two top-level branches (height 2.5). Comparing to the segment numbers shown in Figure 10, it is clear that these represent the river clay landscape, Gelderse Vallei depression, and lower terraces (left branch, e.g., segments 5 and 17) and the sandy uplands (right branch, e.g., segments 1 and 2). At the second level of the right branch (height 1.2) the separation is between small areas with more heterogenous segments (right branch, e.g., segments 4, 6, 9) and the larger more homogeneous areas of the sandy uplands (left branch, e.g., segments 1 and 2). While not a perfect separation, the clustering does separate the principal soil landscape components and their internal heterogeneity.

From an examination of the heights at which groups of segments are joined, it seems that cutting the tree at height 0.8 into five clusters forms the most useful general grouping. This is shown in Figure 13 by boxes around the sets of segments in each cluster. These generalised clusters are shown on the nine properties in Figure 14. They group similar segments well and could serve as landscape management units.

## 375 3.4 Evaluation

By using the algorithms with analyst-selected parameters, the BIS-4D product was able "speak for itself" quite well, revealing both compact units of homogeneous soils and segments with similar heterogeneous patterns of soil classes. Aggregation based on properties and depths selected to represent the results of the principal soil forming factors delineates patches (Figure 5) that closely correspond to polygons of the 1:50 000 design scale conventional soil map with design scale 1:50 000 (Figure 4),

**Figure 12.** Segmentation by whole-profile clay at 1:100 000 (top) and 1:200 000 (bottom) with default thresholds (left) and liberal thresholds (right), overlaid on the clay 0–5 cm DSM product

.

generalized to about 1:100 000 design scale, although with some variations in form. Segmentation was most successful with grid cells of 1 000 ha, corresponding to nominal map scale 1:400 000. This grouped patterns of pixels with different internal patterns of classes. Hierarchical clustering of these segments found groups of similar patterns within the map. These represent separate segments of the same landscape component. These results increase confidence in the BIS-4D DSM product. This is perhaps a best case, due to the high quality of the source data (training points and covariates), the conventional map which can be used for comparison with aggregation and segmentation, and sophisticated modelling approach specific to the Netherlands, as explained by Helfenstein et al. (2024).

Leaves: segment number; Groups: cluster number

**Figure 13.** Hierarchical clustering of the segments shown in Figure 10

## 4 Case Study 2 – SoilGrids v2.0 (Global)

At the other extreme from the country-specific DSM exercise based on a large quality-controlled and spatially complete training set (§3) is a global DSM exercise based on a heterogeneous and spatially-unbalanced training points, using only covariates with global coverage. For this case we selected SoilGrids v2.0 (Poggio et al., 2021) from ISRIC-World Soil Information. This is a set of predictive maps of soil properties for the entire globe at 250 m nominal spatial resolution. Aggregations to 1 km and 5 km resolutions are provided for modelling at coarser scales. It is a globally-consistent product that uses all available point data from the World Soil Information Service (WoSIS) database (Batjes et al., 2024), also from ISRIC-World Soil Information, and covariates with global coverage. Political boundaries are nowhere visible, except where one or more covariates match these. In this it follows the concept of the pioneering FAO-UNESCO Soil Map of the World (FAO - UNESCO, 1971–1979; FAO, 1990).

**Figure 14.** Generalised clusters of the segmentation of Figure 10, based on slicing the clustering dendrogram shown in Figure 13. The same segmentation is shown for nine selected property-depth combinations. Clusters shown by colour and number.

SoilGrids provides both predictions and their uncertainty, via quantile random forest machine-learning models. It closely follows the *GlobalSoilMap* specifications of properties and depth slices (Science Committee, 2015). It also predicts the derived property SOC stocks from 0-30 cm, in T ha<sup>-1</sup>, computed from SOC concentration and bulk density. We selected SOC stock because it is a high priority for global modelling, as evidenced by the efforts of the FAO to produce a global map from national contributions in the Global Soil Organic Carbon Map (GSOCmap) project (FAO, 2018, see a portion of this map in Figure 30, below). It is a high priority due to its key role in soil functions and its importance in policy applications. It is a primary target for DSM over various spatial extents. How can the diverse SOC digital soil maps be evaluated? We propose the spatial pattern and its relation to the soil landscape, as revealed (we hope) by aggregation and segmentation.

Poggio et al. (2021, Table 4) shows that SoilGrids predictions had a median global cross-validation RMSE of 3.97% SOC concentration and  $0.19 \,\mathrm{g \ cm^{-3}}$  bulk density, averaged over the three layers which contribute to SoilGrids SOC stock estimates. We selected a transnational study area with lower-left corner (-109.99 E, 27.86 N) and upper-right corner (-100.03 E, 35.64 N).

Figure 15. SoilGrids v2.0; SOC stock 0-30 cm, T ha<sup>-1</sup>. Test area for aggregation (§4.1) shown as a red square.

This covers most of Chihuahua and Coahuila and part of Sonora States (México) and portions of Texas and New Mexico States (USA). Figure 15 shows this area, with the SOC stocks over the 0–30 cm depth slice. The higher stocks are in mountains and wetlands along the Rio Grande, the lower in high deserts.

Individual 2 × 2° tiles of the 250 m product were downloaded in the Geotiff format from the interactive SoilGrids site (ISRIC-World Soil Information, 2024b), imported into R with the terra package, mosaicked, projected from the original geographic coordinates to a local Albers Equal Area projection, and trimmed to 3 270 x 3 610 6.25 km² pixels, covering 737 793.8 km². The global map of the 1 km product was downloaded in the Geotiff format from the ISRIC WebDAV repository (ISRIC-World Soil Information, 2024a), projected from the original Homolosine coordinate reference system to the same local Albers Equal Area projection, and trimmed to 900 x 900 1 km² pixels, covering 810 000 km².

Figure 16. Test area for aggregation, centred on (-105 E, 32 N). Source: @ Google Earth

Predicted SOC stocks per pixel in the study area ranged from 0 to 83, median  $28 \text{ T ha}^{-1}$  for the 250 m product, and 7 to 76, median  $29 \text{ T ha}^{-1}$  for the 1 km product, showing the smoothing effect of upscaling. These distributions are moderately right-skewed (skewness 0.468 and 0.488, respectively).

## 4.1 Aggregation

- We applied the supercells algorithm to the SOC stocks 250 m resolution layer. To limit processing time and memory requirements, we selected a small test area of 80 x 80 km, i.e., 640 000 ha, centred on (-105 E, 32 N) at the Texas (N) / New Mexico (S) border, near Dell City TX (Figure 16). The centre pivot irrigated fields at the centre-left are  $\approx 800 \times 800$  m and should thus be resolvable on the SoilGrids map. This area includes a wide range of the SOC stocks (Figure 17 left), with high values in the Guadalupe Mountains to the east and very low values in the salt flats in the centre of the area.
- After some experimentation, a medium value (0.5) for *compactness* was selected. We did not set a minimum mappable area minarea, rather a number of proposed supercells  $k \approx 400$  supercells, corresponding to an average area of 1 600 ha and 1 cm<sup>2</sup> on a 1:400 000 map. This is much larger than the area of single centre-pivot irrigated fields, so we did not expect these to be individually resolved.

Figure 17. SoilGrids v2.0 250 m SOC stock 0-30 cm, T ha<sup>-1</sup> (left) and its aggregation into supercells (right).

Figure 17 (right) shows the computed supercells. Median size of the 412 supercells was 1 388 ha, ranging from 431 to 5 462 ha, with a strongly right-skewed distribution. This aggregation clearly groups the pixels with similar SOC concentrations. However, the shapes do not seem to correspond to natural landscape boundaries. We attempted other combinations of compactness and supercell numbers, with poorer results.

The quality of the aggregation can be measured by the standard deviation of the property within each supercell (Figure 18). These ranged from 0.34 to 6.08, median  $1.18 \, \mathrm{T} \, \mathrm{ha}^{-1}$ , with corresponding coefficients of variation from 1.36 to 26.61, median 4.39%. The highest heterogeneity was in the pivot irrigation area, where the minimum supercell size forced pixels with a wide range of values together.

## 4.2 Segmentation

435

Segmentation was applied to the SOC stocks map of the full study area, for both resolution SoilGrids DSM products. Since gpat\_gridhis requires class maps, SOC stocks were classified in 19 (250 m) and 18 (1 km) equal intervals of  $4 \, \mathrm{T} \, \mathrm{ha}^{-1}$ , with from 31 to 1'956 813 (250 m) and 14 to 128 549 (1 km) pixels per class. The minimum grid resolution for the 250 m product is here  $2.5 \times 2.5 \, \mathrm{km}$ . This map was segmented at this resolution, and also four coarser resolutions:  $5 \times 5 \, \mathrm{km}$ ,  $10 \times 10 \, \mathrm{km}$ ,  $20 \times 20 \, \mathrm{km}$ , and  $40 \times 40 \, \mathrm{km}$ , corresponding to map scales 1:1M, 1:2M, 1:4M, 1:8M, and 1:16M, respectively. Table 7 shows the results. As expected, the segments are increasingly heterogeneous as the cell size increases: both the median standard deviation within the segments and their entropy increase.

Figure 19 shows the results of the four finest segmentations. The level of detail is apparent, but many segments at the finest segmentation contain only one SOC class, and thus have no internal pattern. The increasing generalisations find increasing heterogenous segments, with a clearer relation to the soil landscape with each increase.

The 1 km resolution product was also segmented at the four finest possible cell sizes. Again as expected, the segments are increasingly heterogeneous as the cell size increases: both the median standard deviation within the segments and their

Figure 18. Standard deviation within supercells; SoilGrids v2.0 250 m SOC stock 0-30 cm, T ha<sup>-1</sup>, rounded to 0.1 precision

| Cell size      | nominal scale | number of segments | median standard deviation | median normalized entropy | average area km <sup>2</sup> |
|----------------|---------------|--------------------|---------------------------|---------------------------|------------------------------|
| $10 \times 10$ | 1:1'000 000   | 6 612              | 2.36                      | 0.31                      | 122                          |
| $20\times20$   | 1:2'000 000   | 1 718              | 3.07                      | 0.37                      | 471                          |
| $40 \times 40$ | 1:4'000 000   | 485                | 3.86                      | 0.43                      | 1 670                        |
| $80 \times 80$ | 1:8'000 000   | 117                | 4.46                      | 0.47                      | 6 923                        |
| $160\times160$ | 1:16'000 000  | 35                 | 6.42                      | 0.60                      | 23 142                       |

**Table 7.** Results of segmentation of SoilGrids 250 m resolution SOC stock T  $ha^{-1}$ ; normalized entropy [0...1].

entropy increase. Table 8 shows the results. Comparing with Table 7, we see that at comparable nominal resolutions the numbers of segments are comparable, although there are somewhat fewer segments from the 1 km product, consistent with its generalisation.

Figure 20 shows these segmentations of the 1 km product. As minimum segment size increases, broader landscape patterns are increasingly apparent, within the constraint of the rectangular blocks. The coarsest segmentation  $(80 \times 80)$  separates the

**Figure 19.** Segmentation of the SoilGrids v2.0 250 m resolution SOC stock map (part) at (left to right, top to bottom) 1:1M, 1:2M, 1:4M and 1:8M nominal resolutions. Units are  $T ha^{-1}$ 

large low-SOC plateaus from the basin-and-range mountains with alternating high and low SOC. The entire Rio Grande valley is one segment. The next coarsest  $(40 \times 40)$  separates these into segments with somewhat more uniform internal patterns. This resolution will be used for clustering (§4.3, below).

Figure 21 shows the entropy for each segment of the 1:16M nominal resolution map from the 1 km product. This is a measure of the internal class homogeneity of each segment, although not the spatial pattern of the classes. The highest entropies are found in the segments with mixed high and low terrain, shown as contrasting purple and light blue colours.

| Cell size      | nominal scale | number of segments | median standard deviation | median normalized entropy | average area km <sup>2</sup> |
|----------------|---------------|--------------------|---------------------------|---------------------------|------------------------------|
| 10 × 10        | 1:4'000 000   | 581                | 3.44                      | 0.42                      | 1 394                        |
| $20\times20$   | 1:8'000 000   | 151                | 4.51                      | 0.50                      | 5 364                        |
| $40 \times 40$ | 1:16'000 000  | 39                 | 6.08                      | 0.61                      | 20 769                       |
| $80 \times 80$ | 1:32'000 000  | 6                  | 7.98                      | 0.69                      | 135 000                      |

**Table 8.** Results of segmentation of SoilGrids 1km resolution SOC stock  $T ha^{-1}$ 

Figure 20. Segmentation of the SoilGrids v2.0 1 km resolution SOC stock map (part) at (left to right, top to bottom)) 1:2M, 1:4M, 1:8M, and 1:16M nominal resolutions. Units are  $T ha^{-1}$ 

.

**Figure 21.** Normalized Shannon Entropy of segments of the v2.0 1 km resolution SOC stock map (part) at 1:16M nominal resolution. Colour scale from white (lowest entropy) to dark purples (highest entropy).

## 4.3 Clustering

Figure 22 (left) shows the 39 segments signatures from the 1 km product, using motifel size 40 cells, and Figure 22 (right) shows the assignment to seven generalised clusters. Figure 23 shows the dendrogram for the clustering of the 39 segment signatures.

The co-occurrence pattern of classes is similar within each general cluster. The clusters should group similar soil landscapes, at least with respect to the SOC concentration. For example, cluster 1 groups mountainous terrain with high SOC interspersed with basins with medium SOC in an intricate pattern, whereas cluster 5 groups the low-SOC plateau areas. Cluster 2 contains most of the upper Rio Grande valley, but includes some plateau areas to its west.

Figure 24 shows the Jensen-Shannon divergence with the first segment, which necessarily has no divergence. This distance does not directly correspond to cluster distance in the dendrogram unless clustering is by single linkage; here we used clustering by Ward's D2. These range from 0.14 (segment 30, in the same cluster 1 as the target segment, although on a different branch

**Figure 22.** Left: Segmentation of the SoilGrids v2.0 1 km resolution SOC stock map, motifel size 40 cells, units are T ha<sup>-1</sup>; Right: Assignment of segments to five generalised clusters, legend is cluster number.

Figure 23. Dendrogram of segment signatures, SoilGrids v2.0 1 km SOC stock map, motifel size 40 cells, with five general clusters

Figure 24. Jensen-Shannon divergence from Segment 1. Heat colours from red (most similar) to white (least similar).

at height 0.45) to 0.94 (segment 28, in widely-separated cluster 3, different at branch height 1.45). These distances can be used to find the soil patterns that are most similar to any segment, independently of cluster membership.

## 4.4 Evaluation

Aggregation was able to form compact groups of pixels with similar SOC stocks, which could be useful for, e.g., stratified sampling. However the polygons did not seem to correspond to landscape units. Segmentation was more successful. At several increasingly-general scales it grouped distinctive patterns of SOC stocks, corresponding to large landscape units. This was most apparent at the 1:16M nominal resolution (Figure 22). Among the most obvious are the Chihuahuan basin-and-range mountains (segment 29 of Figure 22), the upper Rio Grande valley near Socorro NM (segment 2), and the west Texas/eastern New Mexico plateau (segment 13). Some segments include several physiographic units, which nonetheless apparently had similar patterns of SOC, for example segment 23 which includes some west Texas uplands, the Rio Grande valley below El Paso, and uplands in eastern Chihuahua. Clustering was then able to identify general groups of landscape units, and the Jensen-Shannon divergence identified the segments most similar to a selected segment.

## 5 Case Study 3 – SOLUS100 (USA)






The third case study is intermediate to the first two. Like the BIS-4D study it is of one country and with training points from one source, but (1) it covers a much wider and more diverse area but can't use covariates that are only available for part of the area, and (2) it is based on numerous traditional soil surveys of varying age and quality control, as well as training points, which can be used to some extent for evaluation.

SOLUS100 ("Soil Landscapes of the United States 100-meter") is a recent DSM product from the USDA-NRCS (Nauman et al., 2024). It contains predicted values, high and low estimates, and prediction intervals for soil properties at the *Global-SoilMap* standard depths, at 100 m horizontal resolution (i.e., 1 ha pixels) over the entire conterminous United States (CONUS). The maps are available in GeoTIFF format (Nauman, 2024). SOLUS can be compared to the Gridded Soil Survey Geographic Database (gSSURGO) digital product from the NRCS (NRCS Soils, 2022), which was created by digitising the polygons from traditional soil-landscape survey, with its linked relational database of polygons, map units, components, horizons, and soil properties. NRCS has been working on updates to source maps as well as harmonising map unit names and boundaries across different survey areas since 2013, although this work is not complete. These updates are then used in new versions of gSSURGO. Aggregation and segmentation of SOLUS can be compared to gSSURGO, a product based on expert judgement and field-based soil survey. However, gSSURGO is quite heterogeneous in the age and quality of the soil surveys on which it is based, and so must be approached with caution and preferably with the judgement of a local experienced soil surveyor as to the reliability of gSSURGO.

We selected a 570 km<sup>2</sup> test area in Wayne (Higgins, 1978) and Ontario (Pearson, 1958) Counties NY, originally published in 1978 and 1958 as Order 2, 1:15 840 and 1:20 000 scale surveys, respectively, on an unrectified airphoto base, and later digitised on a topographic base map by the NRCS (D'Avelo and McLeese, 1998) and incorporated into gSSURGO. This area has a distinctive pattern of NNW-SSE orientated drumlins of various sizes and shapes, and inter-drumlin depressions. Some of the depressions developed into peatlands, with drained areas used for agriculture and undrained areas used as wildlife reserves. All soils have developed since the final retreat of the Laurentide Ice Sheet around 12 000 years before present. The main soils are classified in US Soil Taxonomy as Glossic and Oxyaquic Hapludalfs at the tops and sides of the drumlins, and Mollic and Histic Haplaquepts and Medisaprists in the depressions (Soil Survey Staff, 2022). The genesis of this soil landscape has been studied for more than a century (Menzies et al., 2016). A topographic map of a representative portion is shown in Figure 25.

We selected clay concentration and SOC as the properties to analyze. This is because these vary considerably in the area and shows excellent relation with the landscape. Specifically, the inter-drumlin swamps have high SOC and low clay, with the reverse for the drumlins. Accuracy statistics are not available for this area, however, for clay concentration of the 0-5 cm layer over the entire CONUS (Nauman et al., 2024, Table S1) reports spatial cross-validation statistics of 6.481% RMSE, -0.003% ME, and  $0.672~\mathrm{R}^2$ , based on all 484 258 observations. When compared to only the 37 992 observations that were analyzed in the NRCS Soil Characterization Laboratory these results were substantially worse: 8.382% RMSE, 0.011% ME, and 0.544 R<sup>2</sup>. For SOC of the surface layer the statistics are 7.507% RMSE, 0.213% ME, and  $0.716~\mathrm{R}^2$  for all observations, and 4.218%

**Figure 25.** Representative portion of the SOLUS100 test area. Source: USGS topographic map, Lyons NY quadrangle, 2016, scale 1:24 000. Contour interval 10 feet. Projection and marginal coördinates UTM Zone 18N. The centre swamp contains Typic Medisaprists; drumlin tops are Glossic Hapludalfs.

RMSE, 0.062% ME, and  $0.220~\mathrm{R}^2$  for the laboratory observations. Thus the point accuracy of SOLUS for this property is only moderate, but our interest is in the spatial pattern.

Figure 26 shows the predicted surface layer clay concentration for the original soil survey, as compiled in gSSURGO, and for SOLUS. Notice the different legend scales and colour ramps, otherwise the SOLUS map would not clearly show its pattern, since SOLUS predicts a narrower range of concentrations. This is typical of DSM products made with statistical learning methods. (Hastie et al., 2009). It is obvious by visual inspection that SOLUS misses much of the fine pattern, and especially that it does not identify most of the organic soils with very low clay concentrations (dark blue on the gSSURGO map). There is some hint of the pattern in the southeastern corner of the study area.

Figure 26. Clay concentration % of the 0-5 cm layer, gSSURGO (left), SOLUS 100 m (right).

# 525 5.1 Aggregation


Aggregation with the supercells algorithm requires parameterization. We set the minimum area minarea to be comparable to the Minimum Legible Delineation (MLD) (Forbes et al., 1982) at original design scales, 1:15 840 and 1:20 000 for the two counties. We set the reference scale to be a bit smaller, i.e., 1:24 000, so that the MLD was set to 2.304 ha, and increased slightly to three SOLUS cells. Aggregation complexity is also controlled by the target number of supercells. This should be comparable to the number of gSSURGO polygons in this study area. This was to evaluate how well SOLUS in this area can match the traditional soil survey for this property. In this area there are 14 949 gSSURGO polygons, with a median area of 2.43 ha, corresponding very well to the MLD. We reduced this slightly to a target of 14 000 supercells. However, the mean area is 5.30 ha, because of some large polygons of organic soils.

**Figure 27.** Supercells derived from the clay concentration % of the 0-5 cm layer from SOLUS 100 m overlaid by the polygon boundaries (in red) from gSSURGO. Compactness parameter 0.2 (left), 2.0 (right). Projection is UTM18N on WGS84, compare with Figure 25

We first aggregated clay concentration of the surface layer with a range of compactness values from 0.2 to 2. The resulting number of supercells was much lower than the target, ranging from 6 364 for compactness 0.2, to 8 422 for compactness 2.0. As expected, compactness 0.2 produced the map with the most elongated features and 2.0 the least. However the orientation of the supercells did not match the generally NNW-SSE pattern of the drumlin field (Figure 27).

We then aggregated based on clay concentration of all layers, i.e., the full profile, again with a range of compactness. The number of supercells was more consistent than with a single layer, ranging from 7 306 for compactness 0.2, to 8 238 for compactness 2.0. The larger number at the lowest compactness is because the algorithm could not find as much homogeneity in adjacent grid cells when considering all layers. Again, the spatial pattern of the supercells did not resemble the pattern shown by gSSURGO and the topographic map.

From this we conclude that aggregation based on this SOLUS layer does not represent the actual soil pattern. After examining the supercells pattern and the source map, it is unclear to us what the SOLUS model is "seeing" in this area.

## 545 **5.2 Segmentation**



SOLUS resolution is 100 m, so that the minimum shift is 10 i.e., 1 000 m = 1 km, corresponding to 1:250k nominal scale. Thus we did not expect to reproduce the fine pattern, but rather to group these into regions. We segmented with raster stacks of single properties at all depth slices, and with a raster stack of seven properties (clay, silt, and soil organic carbon weight concentrations, coarse fragments volume, pH measured at 1:1 in water, CEC, bulk density) at one depth slice. The continuous properties were converted to classes, as required by the GeoPAT segmentation algorithm: particle-size separates in units of 4%, pH in units of 0.2 pH, CEC in units of  $10 \text{ meq} (100 \text{ g})^{-1}$ , bulk density in units of  $0.1 \text{ kg m}^{-3}$ , and SOC in units of 0.2% up to 6% and then in units of 5% to the maximum of 30%.

**Figure 28.** Segmentation based on all depth slices of SOLUS-predicted SOC concentration, nominal scale 1:400 000, 1000 x %. Note the slightly different colour scales

**Figure 29.** Segmentation based on all depth slices of clay concentration, nominal scale 1:400 000, %. Note the slightly different colour scales.

Figure 28 shows the segmentation based on all depth slices of SOC concentration, overlaid on the concentration at two depth slices. Some segments are well-separated, notably the depressions with swamps and organic soils, as well as sections with different intensities of drumlins. By contrast, Figure 29 shows the segmentation based on all depth slices of clay concentration, overlaid on the concentration at two depth slices. The segments are quite large and do not identify collections of the main landscape elements, i.e., drumlins and depressions.

Similar and even worse results were found with other properties, as well as with an attempt to use all properties at one depth slice.

## 560 5.3 Clustering




Because of the poor results of segmentation, we do not present the results of clustering for this case study.

## 5.4 Evaluation

The two algorithms applied to SOLUS100, with appropriate parameters, allowed the product "speak for itself", but the message was not clear and even misleading. Notably, the attempts to aggregate and segment based on a representation of the profile resulted in unrealistic polygon maps. In this area the landscape pattern is striking and easy to map by conventional methods. SOLUS was unable to approximate the conventional map in this area, let alone improve its resolution. This is likely because SOLUS lacks locally-important covariates to represent this recently-glaciated soil landscape with its characteristic drumlins. This is not meant to be a condemnation of SOLUS as a useful product overall. All DSM models trained over a wide area have difficulty when applied to a local area with idiosyncratic soil-landscape relations which are not reflected in the covariates available over the entire training area, or which have locally-specific relations with the wider-area covariates. This is a general "global model applied to locally-idiosyncratic landscapes" issue, which is being addressed by adaptive methods, see for example Fan et al. (2022). This problem was already recognized early on in DSM exercises. For example Poggio et al. (2010b) discovered that soil available water capacity models used different significant covariates according to the level in a hierarchy of national (Scotland), regional and catchment, and recommended fitting models at the target extent. So in this study area, perhaps fitting the SOLUS model locally would have been more successful in reproducing the soil landscape pattern, even without local covariates related to glaciation.

## 6 Discussion

We first discuss how the two methods performed when applied in the test cases (§6.1), as well as their strengths and limitations, and then discuss how they could be incorporated into evaluating DSM products (§6.2).

## 580 6.1 How did the methods perform?

The supercells algorithm was able to delineate relatively homogeneous soils, based on all soil properties and layers in the BIS-4D example and the SoilGrids SOC example, but failed completely with SOLUS. A limitation of this approach is that there is no objective way to adjust the compactness and supercell number parameters, other than the expert opinion on which choice looks most "realistic". However, the minimum size parameter can be set to match a minimum legible delineation corresponding to a desired map scale.

The GeoPAT algorithm was able to segment DSM products into objectively-defined areas made up of fixed-size blocks, each relatively homogeneous in its pattern internally and relatively isolated from its neighbours. Segmentation was quite successful

on appropriate scales for BIS-4D and the test area and property of SoilGrids v2.0, but much less successful for the test area of SOLUS100. The class composition of segments, although not their internal spatial pattern, were well-characterised by normalized Shannon Entropy.

A limitation of the GeoPAT approach is the requirement for relatively large numbers of pixels per grid cell, and the rectangular shape of the grid cells that are combined into segments. Thus, the segment boundaries can not follow complex natural boundaries. Also, the landscape segments are at much more general scale than the source map.

An obvious question is how to parameterise the two approaches. In this paper we compared several choices of parameters in each case study on an *at hoc* basis. It may be possible to systematise this with sensitivity analysis, to quantify the changes in results as parameters change. This was outside the scope of this paper.

The question remains as to the relation of the supercells or segments with the actual soil landscape at the several scales. There are two related questions. (1) For aggregation, do the relatively homogeneous (according to the supercells algorithm) groups of pixels correspond to landscape elements? These would correspond to polypedons or consociations. (2) For segmentation, do the patterns of pixels within the segment correspond to finer-scale patterns at the design scale of the segmentation? These would correspond to associations or complexes.

In the case of BIS-4D and the detailed traditional Dutch soil survey, the degree to which the aggregation matches the published map (Figure 4) is likely sufficient. The success of segmentation was discussed in §3.2. It is not clear which segmentation scale is the most appropriate.

In the case of SoilGrids, the "true" soil landscape pattern in the test area is not so clear. When comparing SoilGrids with the USA, a problem is that the detailed gSSURGO map (NRCS Soils, 2022) has been compiled from multiple survey areas, mapped over many years, and with imperfect correlation between areas. This is compiled from traditional surveys at design scales from 1:12 000 to 1:24 000 in most areas, but somewhat coarser in less populated areas in the western USA. The INEGI map in México is a consistent 1:250'000 national product (Instituto Nacional de Estadística, Geografía e Informática (INEGI), 2024), which can show a minimum delineation of 250 ha. Figure 30 shows a SOC stock maps of the study area, compiled from the above-mentioned USA and Mexican sources by the FAO as part of the Global Soil Organic Carbon Map (GSOCmap) project (FAO, 2018). Version 1.6.1 of this product was downloaded from the FAO's Global Soil Information System (GloSIS) (FAO, 2024). The inconsistency in values and pattern between México and the USA is obvious, as are several sharp boundaries between survey areas in the USA. So it is difficult to evaluate how well SoilGrids identifies supercells or segments.

In the SOLUS example, the geomorphology and soil pattern of the test area is well understood and has been mapped in detail. Of the SOLUS layers only soil organic carbon and coarse fragment volume showed a relation with known patterns in the test area. Aggregation based on multiple properties completely failed to find landscape units. Segmentation based on multiple properties failed to find more general units with consistent internal patterns.

## 6.2 Evaluating a DSM product







So, how should aggregation and segmentation be used in an overall evaluation of a DSM product? The common use of point evaluation statistics by cross-validation or repeated data splitting is still important, as long as the representativeness in both

**Figure 30.** Global Soil Organic Carbon (GSOC) map (part). Boundary is between México (south) and the USA (north). Values are T ha<sup>-1</sup> SOC stock. Source and legend: FAO (2024).

geographic and feature space is clear to the map user. There is a large difference between these statistics applied to legacy observations that were opportunistically located (e.g., farmer-supplied observations), purposively located (e.g., at "typical" locations for soil series), or placed by a method meant to cover feature space, e.g., conditioned Latin hypercube sampling (Minasny and McBratney, 2006) or geographic space, e.g., spatial coverage sampling Walvoort et al. (2009). But as explained in the Introduction (§1), these do not account for spatial patterns.




An obvious evaluation of aggregation and segmentation can be the expert opinion of the soil geographer familiar with the mapped area. Notable soil-landscape features should be identified either by aggregation for relatively homogeneous areas such as swamps and salt flats, or segmentation for heterogeneous areas at the design scale, for example prairie pothole topography (Kiss et al., 2022) at scales where individual potholes can not be shown. Although subjective, this can be supported by the geographer's conceptual model (Hudson, 1992) based on field experience and known landscape expression. An example is the discussion of the SOLUS map in the well-understood soil landscape of Case Study 3 (§5). All soil surveyors and most field scientists using soil maps soon recognise that some conventional maps are more reliable than others, that is, some delineations are more reliably identified than others. So just matching a conventional map at the appropriate degree of generalisation is not always appropriate. In Case Study 3 the landscape and soil patterns are highly distinctive so that the original surveyors could hardly make mistakes – the only problem could be digitizing from the unrectified photo base used for the original survey to a correct topographic base for incorporation in SSURGO. In other contexts the soil-landscape relation and soil boundaries may not be so clear and so difficult to represent (Lagacherie et al., 1996), and in others the conventional map may have been made

by less-skilled surveyors. No general solution can be given – this is a separate level of expert opinion, i.e., the reliability of the traditional map.

The starting point in any evaluation is the intended use(s) of the map. Then its fitness for use can be assessed according to the requirements to support those uses. Pattern-based evaluation is indicated if be map be used to represent soil geography, for example, to help map users assess the relation of soils with the landscape. It is also indicated if the map user will need to identify landscape components, for example for ecological zoning of a protected area. The degree of internal heterogeneity as revealed by the segmentation can be used to assess connectivity, for example in catchment hydrological models.

One application where segmentation analysis can be used is identifying areas similar in their internal spatial pattern to a known area where the pattern has been characterized. This has been applied to land cover (Nowosad, 2018) but can apply equally well to soil patterns. For example, in every region there are areas with high sampling density and well-characterized soils, and others with less information. Once the segments are established over the whole area, specific segments in the high-density area can be matched to those in the low-density area, where the soil pattern is expected to be similar. This is the "Homosoil" concept (Nenkam et al., 2022) applied to areas. The clustering of segments in the BIS-4D (§3) and SoilGrids (§4) case studies shows one way to do this. The Jensen-Shannon distance from a target segment can also be used to identify the most similar segments.

#### 7 Conclusions






The methods presented in this paper are part of an effort to evaluate DSM products based on how well they represent the soil landscape. The approach taken here complements pattern analysis of the DSM product, which characterises the map without attempting aggregation or segmentation, as in Rossiter et al. (2022).

Both the aggregation and segmentation approaches were able to allow the DSM product "speak for itself", with the assistance of the analyst's choices of parameters. Individual predictions in pixels were combined into possible soil-landscape elements, which could be evaluated statistically and by expert judgment. Both of these approaches require the intervention of the analyst to select scales and parameters, often with large differences in resulting patterns. This has the advantage that the analyst can match desired scales of landscape analysis, and indeed can perform a multi-resolution evaluation. The analysis of the resulting maps is a significant addition to the commonly-used "point"-based evaluation statistics, which (1) do not evaluate the full map, (2) even at point support, do not take into account the spatial relation between evaluation points. We hope that this will stimulate digital soil mappers to evaluate their own products in this light. This should lead to clearer communication with DSM users, so that digital soil maps become more widely accepted and properly used.

Code and data availability. The GeoPAT modules are available at its GitHub repository<sup>1</sup>. The superpixels R package is available at  $CRAN^2$  and must be installed from within the R environment. The analysis code for this paper is available in a GitLab repository<sup>3</sup>. The datasets used in case studies can be obtained from the websites referenced in the text.

*Author contributions*. DGR conceived of the approach, wrote the code, carried out the test cases, and wrote the initial draft. LP contributed advice at every step, especially the concepts and interpretations, and contributed detailed knowledge of SoilGrids.

Competing interests. The authors have no competing interests

<sup>&</sup>lt;sup>1</sup>https://github.com/Nowosad/geopat2

<sup>&</sup>lt;sup>2</sup>https://cran.r-project.org/web/packages/supercells/index.html

<sup>&</sup>lt;sup>3</sup>https://git.wur.nl/isric/scientific-publications/Rossiter-2025-Soil\_landscapes\_from\_DSM

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
