# Peer review of "Representing soil landscapes from digital soil mapping products – helping the map to speak for itself"

_EGUsphere, 2025_

## Referee Comment (RC2)

Review paper: egusphere-2025-1896

General View of the Paper:

This paper introduces and applies a pattern-based evaluation approach for Digital Soil Mapping (DSM) products, built around the concept of letting "the map speak for itself." Through three distinct case studies—BIS-4D in the Netherlands, the global SoilGrids v2.0 product, and SOLUS100 in the USA—the authors explore how aggregation, segmentation, and clustering methods can be used to evaluate whether DSM products reflect meaningful soil landscape patterns. This approach complements traditional point-based accuracy assessments by focusing instead on spatial coherence and landscape-level pattern recognition.

One of the strengths of this paper is that it goes beyond traditional point-based accuracy metrics and puts emphasis on evaluating the **spatial realism** of DSM outputs. This aspect is often neglected in many DSM studies, where maps are validated statistically, but not checked visually or structurally to see if they actually reflect real soil landscape patterns.

The case studies are carefully chosen to illustrate the strengths and limitations of the methods under different mapping conditions: one that performs well (BIS-4D), one global product with mixed success (SoilGrids), and one that fails to capture the landscape structure (SOLUS). This comparative structure strengthens the paper and offers valuable insights for practitioners working with DSM.

That said, the paper can be quite dense and difficult to follow in places due to the high level of technical detail. Long descriptive passages filled with numeric results might be better supported by well-structured summary tables. For example, in the Netherlands case study, a table showing different properties, segmentation scales, and the resulting number or size of segments would allow readers to quickly scan and compare outcomes across scenarios. Presenting complex results visually or in tabular form would improve readability and accessibility.

Figures are generally effective, Figure 11 (left), for instance, is particularly compelling, showing detailed patch delineation that appears well aligned with conventional maps. This kind of visual confirmation is rarely provided in DSM studies, yet it plays a crucial role in demonstrating the practical utility of such products. The paper succeeds in making a strong case for including spatial pattern analysis in DSM evaluation workflows.

In summary, the study is a valuable contribution to the DSM literature, highlighting a needed shift toward landscape-aware evaluation. It will be especially useful for practitioners seeking to validate their maps beyond traditional metrics and foster broader acceptance of DSM by bridging the gap between digital outputs and expert expectations from conventional mapping.

I would recommend this paper for publication in *EGU Sphere*, pending minor adjustments. Please find my detailed comments below:

**Introduction**

**General terminology:** Readers outside soil geography may be unfamiliar with specialized terms such as *supercells*, *consociation*, and *SCORPAN*. Please provide a brief parenthetical definition (or a short glossary) on first mention of each term.

**Line 40:** The phrase *"catena of Milne"* could confuse readers who do not know the historical context. Add the publication year to anchor the historical reference.

Consider wording such as:

"…the classic example is the *catena*, as defined by Milne (1935), meaning 'a sequence of distinct but pedogenetically related soils consistently located on specific slope facets…' (Borden et al., 2020)."

This makes it explicit that "Milne" refers to Geoffrey Milne's original definition of a catena.

**Line 50**: "At increasingly detailed scales and with increasingly fine distinctions in the definition of soil bodies, increasingly finer patterns are revealed" This sentence is conceptually clear but stylistically repetitive. The repeated use of "increasingly" and "fine" makes it feel redundant and slightly difficult to follow. I suggest rephrasing it for clarity and flow.

**Line 97:** The objectives paragraph states that the study will derive soil-landscape units via aggregation, segmentation, and clustering, and that these units can be used for routine DSM evaluation. However, it is unclear **how** the derived patterns will be incorporated into the evaluation itself (e.g., spatial correspondence to legacy polygon maps, pattern-matching indices, error aggregation by unit). A brief statement of the specific pattern-based metrics or comparison framework you intend to use would help readers see exactly how the new units feed into DSM evaluation.

**Materials and Methods:**

**Line 105**: Typo – "Aggregnation" should be corrected to aggregation.

**Line 109**: Typo – Remove the duplicate "in" in "defined in in multivariate space."

**Line 109**: Remove the stray period after the Lin (1991) citation.

**Line 125**: The sentence "Unlike supercells, segments must have rectilinear borders" could benefit from a brief explanation of the cartographic rationale behind this constraint. Why is this a requirement?

**Figure 2**: Please spell out the abbreviations "gpat_gridhis" and "gpat_gridts" in the caption (and first mention in the text), so readers immediately understand what each tool does within the workflow.

**Line 143**: For smoother readability, rephrase to: "Two important thresholds for joining grid cells into segments are:"

**Line 146**: "Spatial" does not need to be capitalized here.

**Lines 175 - 177**: Avoid using the vague term "better." Clarify what "better" means in this context, does it refer to lower inhomogeneity, higher isolation, or both? Clearly define the direction of desired improvement for each metric to avoid ambiguity.

**Figure 3**: Again, ensure that abbreviations like "gpat_polygons" and "gpat_distmtx" are spelled out in the caption, especially for readers unfamiliar with the GeoPAT suite.

Consistency issue: You shift between past and present tense ("Here we used…" vs. "Here we use…"). Choose either past or present tense and apply it consistently throughout the **Methods** section.

**Case studies:**
**Case study 1 – BIS-4D (Netherlands)**
**Line 205**: 54 maps (7 properties, 6 layers):  fine, though "depth layers" may be clearer than "layers".
**Line 211**: minarea was to 1 600 25 m x 25 m pixels" → typo → should be "minarea was set to 1,600 pixels (25 m × 25 m).
**Line 221**: classification of soil property maps for example: pH (0.1), clay (5 %), etc., are arbitrary to the reader and not clear enough how you choose these values. Provide a brief justification (measurement    precision,    agronomic    relevance,    histogram    breakpoints).
**Line 264**: isn't it 54 layers?
**Line 265**: It is not clear which depth for which soil properties has been used, rewrite the whole sentence: for example, pH, clay, silt and SOM for the depth of 0-5 cm, clay and bulk density for the depth of 15-30 cm and continue. Otherwise, the reader would be confused.
**Line 281**: As you mentioned it, segmentation is greatly affected by two thresholds, it doesn't seem that default thresholds and liberal threshold would produce comparable maps. As I don't see the segments capture the distinct pattern in figure 11 (right) with more liberal thresholds.
**Section 3.3** Clustering: First paragraph: It is not clear how to follow cluster numbers? And how you link each cluster to the soil landscape components? But also state how the number of clusters is selected for this case.
**Line 305**: "due to the extremely high quality", it would be nice to add the quality of source data for better comparison.

In general, it would strengthen the paper to include basic descriptive statistics and accuracy/error metrics for the DSM products or baseline maps used in each case study. For example, if the maps are described as having "good accuracy," providing supporting metrics—such as RMSE, $R^2$, or standard deviation for selected properties (e.g., pH)—would give readers a clearer understanding of the underlying data quality. These statistics could be included in the Supplementary Materials if space is a concern, but they would offer helpful context when interpreting the segmentation and aggregation results.

**Case study 2 – SoilGrids v2.0 (Global)**
**Line 338**: mineara" → typo → should be **minarea**
**Line 356:** "many segments seem to be of a single class" → vague. Maybe clarify in this way: "...many segments contain only one SOC class, offering little internal variability."

**Figure 17**: the highest standard deviation appears to be approximately **6.1**, but in the text (Line 346), it is stated that the standard deviation ranges from 0.34 to **6.08**. Please double-check this value, there may be a small discrepancy between the figure and the text.

**Section 4.4**: While Section 4.4 offers a qualitative summary of the results from aggregation, segmentation, and clustering, it remains unclear how these steps contribute to the evaluation of DSM products, which is a central stated goal of the paper. The discussion is primarily descriptive and does not explain whether the derived spatial units are being used to validate DSM predictions, identify mapping artifacts, support field design, or quantify model accuracy. To strengthen the study's impact, I recommend adding a brief clarification of the intended **evaluation framework**— for example, whether it involves comparison to legacy soil maps, pattern-based quality metrics, or expert validation. This would help readers understand how these methods serve as tools for DSM assessment rather than just spatial analysis outputs.

**Case study 3 – SOLUS100 (USA)**
**Line 419**: The mean is area 5.30 ha -> the mean area is
**Line 424**: The statement "From this we conclude that SOLUS in no way represents the actual soil pattern*"* may be too strong without formal pattern-matching metrics or more comprehensive property comparisons. Consider softening this claim or supporting it with a visual or statistical metric beyond visual inspection.

The authors clearly demonstrate the challenges of applying the DSM approach of SOLUS100 in a glaciated landscape with fine-scale patterning. Despite methodical application of aggregation and segmentation, the results did not reflect landscape features accurately, suggesting a mismatch between the DSM product's input data and the local soil-forming processes.

**Discussion:**
The authors acknowledge limitations in parameter selection, but what guidance can they offer for users who are not yet experts? This study suggests that appropriate parameter selection is crucial for success, otherwise, the methods may fail to produce meaningful results. For instance, the text states, "There is no objective way to adjust compactness and supercell number parameters." It would be helpful if the authors could recommend whether any metrics (e.g., internal variability, boundary length, or other summary statistics) might serve as semi-objective tools for parameter tuning, even as part of a sensitivity analysis.

The discussion refers to traditional mapping units such as polypedons, consociations, complexes, and associations. It would be helpful if the authors clarified whether their derived units are intended to correspond to these traditional concepts, or whether they serve a different function within the context of DSM evaluation. As the authors note, traditional surveys themselves can be inconsistent—particularly when comparing products like gSSURGO in the U.S. and the INEGI map in Mexico. This raises the question of which map should be treated as the baseline for comparison, especially when these traditional sources vary in quality and methodology.

**Conclusion:**

The conclusion effectively summarizes the intent and broader value of the methods, particularly the shift toward spatial pattern-based evaluation of DSM products. However, the claim that the map is allowed to "speak for itself" might be reconsidered or clarified, since the process still depends heavily on analyst-defined parameters.

---

## Referee Comment (RC3)

**Representing soil landscapes from digital soil mapping products – let the map speak for itself**

David G. Rossiter[1,2] and Laura Poggio[1]

[1]ISRIC-World Soil Information, Wageningen (NL)
[2]Section of Soil & Crop Sciences, College of Agriculture & Life Sciences, Cornell University, Ithaca NY 14850 (USA)

**Correspondence:** David G. Rossiter (david.rossiter@isric.org)

* using vars + depths ... space vs. depth variance ...
    ↳ does this generalize across
       "simple" + "complex" horizonation?

[revised manuscript text omitted]

Together, these make up the soilscape, i.e., distribution of polypedons on the landscape. These form a pattern. The classic 40 example is the catena of Milne: "a sequence of distinct but pedogenetically-related soils that are consistently located on specific facets down a slope, giving recurrent topographically associated soil pattern" (Borden et al., 2020). We would hope that a map of a catena would clearly show these elements and their transitions.

In traditional, expert-based soil class mapping (Hudson, 1992) the landscape is segmented according to the mapper's conceptual model of soil-landscape relations, and by examination of external clues, notably relief, vegetation, and land use, and 45 by augering or full profile examination. DSM replaces the conceptual model with correlative relations with digital coverages meant to represent, at least in part, one or more of the seven predictive SCORPAN factors of McBratney et al. (2003). Therefore, there is no longer an explicit relation with the soil landscape, but it is hoped that the implicit correlative relations can find these.

The concept of areas with distinct patterns of contrasting soils goes back to the "soilscape fabrics" from the soilscape 50 analysis of Hole (1978) and the "soil combinations" of Fridland (1974). At increasingly detailed scales and with increasingly fine distinctions in the definition of soil bodies, increasingly finer patterns are revealed. Conversely, at coarser scales, patterns are based on less precise definitions of distinct soil bodies. As Fridland puts it, "Soil combinations consist of elementary soil

[Figure]

[Figure]

**Figure 1.** Conceptual block diagram, Otsego County NY (USA)

Source: https://www.nrcs.usda.gov/publications/NY-2010-09-28-14.png

[revised manuscript text omitted]

Thus, the objective of this study is present methods to create presumed soil landscape units from DSM products, by both aggregation and segmentation, and then to cluster the segments to identify similar soil landscape units within the map. We first describe the methods and then apply them to three case studies corresponding to different DSM projects at various resolutions and extents. Finally, we discuss how these methods can be used in routine evaluation of DSM products.

*[handwritten: This could use a more detailed definition + target "scale": a spatial + conceptual ...]*

[Figure]

**2 Methods**

We contrast two approaches to letting the map "speak for itself": aggregation based on homogeneity of properties (§2.1), and segmentation based on patterns of classified properties within segments (§2.2).

**2.1 Aggregation**

Aggregnation seeks to find contiguous groups of pixels with relatively homogeneous property values, either single or multivariate. This is implemented by the `supercells` R package (Nowosad, 2025), which uses the Simple Linear Iterative Clustering (SLIC) image-processing algorithm (Nowosad and Stepinski, 2022), with the improvement that an appropriate data distance measure and function for cluster averaging can be defined. For multivariate aggregation there must be a distance measure defined in in multivariate space. A common choice, used here, is the Jensen-Shannon divergence, (Lin, 1991), which quantifies the distance between two histograms by the deviation between the Shannon entropy of the combination of two uni- or multivariate histograms and the mean of their individual entropies.

The `supercells` function is controlled by several parameters that have a large effect on the results. First and most important is *compactness*, which trades off internal homogeneity of the supercells with their geometric compactness. The absolute compactness value depends on the range of input pixel values and the selected distance measure. A large value prioritizes spatial distances between pixels and superpixel centres (more geometric compactness), whereas a smaller value prioritizes distances in feature space (more property homogeneity). Second is the approximate number of supercells, $k$. This should correspond to the number of landscape segments expected in the study area, at the design scale of the corresponding polygon map. Third is the minimum supercell size, *minarea*. This should correspond to a minimum mappable area or a minimum size needed for an application, e.g., land management or stratified sampling.

The quality of the aggregation can be evaluated by the standard deviation or coefficient of variability of each property in the supercell. As supercells decrease in size, these measures will necessarily have smaller values.

**2.2 Segmentation**

Segmentation seeks to find contiguous groups of blocks of grid cells with similar internal patterns of pixels, which represent soil classes or properties, these either univariate or multivariate. Patterns are computed within blocks of at least 10 x 10 pixels, as specified by the analyst. Unlike supercells, segments must have rectilinear borders.

[revised manuscript text omitted]
 $[0\ldots1]$ scale and is stable. We chose Ward's with squared distances to minimize within-cluster variance.

*explain please*

**3 Case study 1 – BIS-4D (Netherlands)**

190 BIS-4D ("Bodeninformatiesysteem 4-Dimensional") (Helfenstein et al., 2024) is a high-resolution (25 m horizontal, six depth slices vertical) soil modelling and mapping platform for the Netherlands. The 3D are geographic space and depth along the soil profile. The fourth dimension is time, applied only to soil organic matter (SOM), which we ignore here by using only the most recent SOM map. Predicted properties are clay, silt, sand and SOM concentrations %, bulk density $g\,cm^{-3}$, pH in KCl, total N $mg\,kg^{-1}$, oxalate-extractable P $mmol\,kg^{-1}$, and cation exchange capacity $mmol(c)\,kg^{-1}$. Depth slices are

195 the *GlobalSoilMap* standard 0–5, 5–15, 15–30, 30–60, 60–100 and 100-200 cm (Science Committee, 2015). Each map is

[Figure]

[Figure]

**Figure 4.** Semi-detailed soil map of the Netherlands, design scale 1:50 000 (part).

Source and detailed legend: Ministerie van Volkshuisvesting en Ruimtelijke Ordening (2024).

General legend: Dark and medium green: river clays with different clay concentrations; Light green: glacial depression sediments; Brown, pink: push moraines with varying sand and gravel sizes; Yellow: wind-blown sands; Purple: peat.

accompanied by uncertainties (quantiles and 90% prediction interval). We did not use these in this analysis, only the mean predictions. Coverages in the *GeoTIFF* format are free to download and use, and can be directly read into the `terra` R package (Hijmans et al., 2025).

BIS-4D is highly accurate at point support, as assessed by cross-validation, due to a very dense sampling network and the
200   country-specific covariates used in the DSM. Visual inspection of layers agrees well with traditional 1:50 000 scale polygon soil maps (Steur and Heijink, 1980; Brouwer et al., 2021) and expert views of the soil landscape.

We selected a 40 x 40 km test area (Figure 4), because of its diverse soil-forming environments, including river clays of various ages and compositions, sandy push moraines, organic soils in glacial depressions, and coversands.

[Figure]

**3.1 Aggregation**

205 The `supercells` algorithm can work directly on raster stacks of the `terra` package. All 54 maps (7 properties, 6 layers) were combined in a `SpatRaster` raster stack. Since the values and ranges are not compatible, the Jensen-Shannon divergence was used to evaluate the distance in feature space between pixels and supercell centres. In this landscape there are non-compact (extended) features parallel to the river, in the fen areas and along the push moraines, so after some experimentation a low *compactness* value (0.2) was selected. We selected a minimum mappable area of 10 ha, equivalent to the 1:50 000 design scale

210 of the Dutch conventional soil map, using the Cornell definition of 0.4 cm$^2$ minimum legible area on the map (Forbes et al., 1982). Thus we set the *minarea* was to 1,600 25 m x 25 m pixels. *typo? missing comma?*

Figure 5 shows the supercells (outlined in black) with several properties as a background. Note that the supercells in all maps are the same, but of course the mean values of each property within the supercells are different. The median size of the 270 supercells was 433 ha, ranging from 104 to 5 044 ha, with a strongly right-skewed distribution. Aggregation clearly shows

215 the differences between soil bodies, with some properties being more prominent in certain supercells.

To evaluate the quality of the aggregation, we computed the standard deviation of each property within each supercell (Figure 6). These are quite low for clay and SOM, and for pH with some small but notable exceptions. Bulk density is less successfully aggregated. The exceptions are where that property is not important in the computation of Jensen-Shannon divergence to that supercell. *Explain*

220 ## 3.2 Segmentation

Since `gpat_gridhis` requires class maps, we classified the soil property maps as follows: bulk density by 0.1 g cm$^{-3}$, CEC by 25 mmol(c) kg$^{-1}$, clay, silt, sand concentrations by 5%, P$_{ox}$ by 4 mmol kg$^{-1}$, pH by 0.1 units, SOM concentration by 4%, and total N by 1000 mg kg$^{-1}$.

The minimum grid size for segmentation (10 x 10 pixels) is 250 x 250 m (62.5 ha), corresponding to a 1:158 000 scale map,

225 as explained in §2.2. Segmentation at this resolution is expected to more closely match the 1:200 000 generalised soil map (Haans, 1965) than the 1:50 000 semi-detailed map shown in Figure 3.

**3.2.1 Univariate segmentation of individual maps**

To examine the effect of grid size, we segmented all properties at all depths, individually, at the minimum possible grid cell size, i.e., 10 × 10 and at several multiples: 40 × 40 (1 000 ha) and 80 × 80 (4 000 ha), corresponding to nominal map scales

230 1:400,000 and 1:800,000, respectively. *commas?*

The finest segmentation produced 4,393 (pH 100–200 cm) to 675 (Pox 100–200 cm), median 2,678 segments, average area 0.597 km$^2$. Comparing this to the single grid cell at resolution, 0.625 km$^2$, we see that many segments were of one or two grid cells. The pattern was mostly very fine, with a few large segments for most single properties.

Segmentation at 1:400,000 equivalent scale produced 231 (sand 0–15 cm) to 41 (SOM 15–30 cm), median 181 segments,

235 average area 8.84 km$^2$. Compared to the single grid cell at resolution, i.e., 1 km$^2$, there was significant grouping. Segmenta-

[Figure]

[Figure]

[Figure]

**Figure 5.** Results for selected properties of aggregation by `supercells` algorithm using all properties and layers

tion at 1:800 000 equivalent scale produced 66 (sand 0–15 cm) to 12 (Pox 60–100 cm), median 47 segments, average area $34.04 \, \text{km}^2$. Again, compared to the single grid cell at resolution, i.e., $4 \, \text{km}^2$, there was significant grouping.

**3.2.2 Multivariate segmentation of individual properties, all depth slices**

We then performed a multivariate segmentation using all depth slices of single properties. By default, GeoPAT normalizes each
240    layer and by default weights them equally. In this mode, a motifel must meet the threshold conditions for all input layers to be joined to a segment. In this way the segmentation is meaningful for each layer. Because of the different spatial structures of the properties at each depth slice, it was expected that the segmentation would be finer at each scale than for individual depth slices, i.e., it would be more difficult to merge grid cells.

The finest segmentation using all depth slices of a single property produced 3 316 (pH) to 168 (SOM) segments, median 1 873
245    segments, average area $0.854 \, \text{km}^2$. Segmentation at 1:400 000 equivalent scale produced 190 (sand) to 13 (SOM) segments, median 127 segments, average area $12.59 \, \text{km}^2$. Segmentation at 1:800 000 equivalent scale produced 55 (sand) to 6 (SOM)

[Figure]

[Figure]

**Figure 6.** Standard deviations for selected properties of aggregation by `supercells` algorithm using all properties and layers

segments, median 36 segments, average area 44.44 km$^2$. Contrary to our expectations, the median number of segments were all smaller than those for the corresponding property's single depth slice segmentations.

Figure 7 shows the segment boundaries for this multivariate segmentation by bulk density over the whole profile, at the three resolutions overlaid on the Dutch 1:50 000 soil survey polygons. It is clear that the 1:800 000 segmentation misses important differences and that the 1:100 000 segmentation finds quite small areas, mostly just one grid cell, within soil bodies. The 1:400 000 segmentation (i.e., shift size 40, 1 km$^2$) grid cells) matches well with many soil map boundaries.

Figure 8 shows the success of the segmentation based on bulk density over the whole profile at the 1:400 000 design scale: inhomogeneity of each segment and isolation from its neighbours. For example, the pixels in the large segment in the top-centre are quite similar in their bulk density profiles, but this segment is only moderately different from its neighbours. This shows the relative homogeneity of the bulk density profiles of the central Gelderse Vallei (Gelderland Valley) in the vicinity of Renswoude and Scherpenzeel. Note that this area also has large segments based on all properties and depth slices, as seen in Figure 5.

[Figure]

[Figure]

*Can these be larger? Hard to "see" patterns*

[Figure]

**Figure 7.** Segmentation based on bulk density over the whole profile (red lines), overlaid on soil map polygons (grey lines). Design scales left to right: 1:100 000, 1:400 000, 1:800 000

*How can we interpret these figures?*

[Figure]

**Figure 8.** Evaluation of segmentation based on bulk density over the whole profile at the 1:400 000 design scale

As the segmentation becomes coarser the inhomogeneity and isolation both decrease, i.e., segments are internally more consistent in their patterns, and less isolated from their neighbours. For example, median inhomogeneity values from the segmentation based on whole-profile bulk density (1 266, 96, 28 segments) decreased from 0.108, 0.086, to 0.076. In parallel, median isolation values decreased from 0.288, 0.211, to 0.178.

**3.2.3 Multivariate segmentation with selected properties and depth slices**

Another segmentation is obtained by selecting properties and depth slices to represent the profile. Using all 56 layers results in an impractical Jensen-Shannon divergence, hence we selected key properties at key depths: pH, clay, silt, SOM 0-5 cm, clay, bulk density 15-30 cm, CEC 30-60 cm, sand, SOM 100-200. Figure 9 shows the segment boundaries from this segmentation at the 1:400 000 design scale, overlaid on several single soil properties and depth slices. Note that the segment boundaries are the same for all maps. This segmentation should best group soils considered holistically, not per-property. Many of the segments

*Why?*

[Figure]

[Figure]

[Figure]

**Figure 9.** Segmentation based on selected properties and depth slices, overlaid on DSM of selected soil properties, 1:400 000 design scale. Legends not shown. Scale is from dark red (low values of the property) to dark green (high values). Top-left map includes segment numbers

correspond to landscape features shown in the conventional soil map of Figure 4, although constrained to the rectilinear shape
270   and minimum grid cell size.

**3.2.4   Scaling of segmentation**

*— meaning? efficient? / "generalizes well"?*

The segmentation method scales well. The land area of the Netherlands ($\approx 33\,240\ \text{km}^2$) was segmented using all depth slices
for several properties. At the nominal 1:400 000 design scale, this resulted in 2 535 (pH) and 1 547 (bulk density) segments; at
1:800 000 design scale 649 (pH) and 371 (bulk density). Figure 10 shows the segmentation by pH of the entire Netherlands at
275   these two scales. For this extent the coarsest segmentation seems most useful for understanding the country-wide soil pattern.

[Figure]

[Figure]

[Figure]

[Figure]

**Figure 10.** Segmentation by whole-profile pH of the Netherlands at 1:400 000 (left) and 1:800 000 (right) nominal scales, overlaid on the pH 15–30 cm DSM product

**3.2.5   Segmentation parameters**

Segmentation is greatly affected by the two thresholds. For example, segmenting the test area using all depth slices for clay using the default lower and upper thresholds (0.1 and 0.3, respectively) results in 1 932 (1:100 000) and 148 (1:400 000) segments, whereas using more liberal (easier segmentation) thresholds 0.3 and 0.8 the number of segments is reduced to 285 and 18. In effect, the more liberal segmentation at a finer scale is similar to the more conservative one at a coarser scale. Figure 11 shows the multivariate segmentation of the test area on the basis of clay concentration at all depth slices at nominal 1:400 000 scale with default thresholds, and the same for the 1:100 000 scale but with liberal thresholds. These maps are comparable.

**3.3   Clustering   *helpful interpretation**

Hierarchical clustering was applied to the segments of Figure 9, i.e., based on selected properties and depth slices, to represent the profile. The resulting dendrogram is shown in Figure 12. Note the large separation in internal patterns between the two top-level branches (height 6). These represent the river clay landscape, Gelderse Vallei depression, and lower terraces (right branch, clusters 4–7) and the sandy uplands (left, clusters 1–3). At the second level for the right branch (height 3.5) the large separation is between the Gelderse Vallei depression and terraces (clusters 4 and 5) and the river clays (clusters 6–7). At the third level for the rightmost branch is the separation between the actively flooded zones (cluster 7) and the somewhat higher zones (cluster 6). While not a perfect separation, the clustering does separate the principal soil landscape components.

The seven generalised clusters identified in the dendrogram are shown on the landscape in Figure 13. These group similar segments well and could serve as landscape management units. For example, cluster 4 groups the mostly homogeneous seg-

[Figure]

*black lines hard to "see"
consider alt. color ramp + lines*

[Figure]

**Figure 11.** Segmentation by whole-profile clay at 1:400 000 with default thresholds (left) and 1:100 000 (right) with liberal thresholds, overlaid on the clay 0–5 cm DSM product

*— worth comparing to multivariate segmentation*

ments dominated by low pH, clay, SOM, CEC, high sand, and medium silt. Cluster 7 groups the heterogeneous segments along
295 the rivers and large brooks.

**3.4 Evaluation**

The BIS-4D product can "speak for itself" quite well, to reveal both compact units of homogeneous soils and segments with similar heterogeneous patterns of soil classes. Aggregation based on properties and depths selected to represent the results of the principal soil forming factors delineates patches (Figures 5 and 9) that closely correspond to polygons of the 1:50 000 design
300 scale conventional soil map with design scale 1:50 000 (Figure 4), generalized to about 1:158 000 design scale, although with some variations in form. Segmentation was most successful with grid cells of 1 000 ha, corresponding to nominal map scale 1:400 000. This grouped patterns of pixels with different internal patterns of classes. Hierarchical clustering of these segments found groups of similar patterns within the map. These represent separate segments of the same landscape component. These results increase confidence in the BIS-4D DSM product. This is perhaps a best case, due to the extremely high quality of the
305 source data (training points and covariates), the conventional map which can be used for comparison with aggregation and segmentation, and sophisticated modelling approach specific to the Netherlands.

**4 Case study 2 – SoilGrids v2.0 (Global)**

At the other extreme from the country-specific DSM exercise based on a large quality-controlled and spatially complete training set (§3) is a global DSM exercise based on a heterogeneous and spatially-unbalanced training points, using only covariates with
310 global coverage. For this case we selected SoilGrids v2.0 (Poggio et al., 2021) from ISRIC-World Soil Information. This is a

[Figure]

*suggestions :*
- *horizontal layout w/ ape package*
- *use height scale + annotations to follow text in 3.3*
-  *label clusters w/ numbers*
- *if not referencing segment ID, do not include*
- *use shaded symbols/filled symbols vs. colored outlines*
  - → *place filled symbols @ terminal "tips" of dendrogram*

*← mentioned in text: mark for readers.*

*color is too hard to "see"*

**Figure 12.** Hierarchical clustering of the segments shown in Figure 9

*e.g. { 1*

*2    3    4    7    6    5*

*annotate groups w/ interpretation*

*e.g. "river clays"*

set of predictive maps of soil properties for the entire globe at 250 m nominal spatial resolution. Aggregations to 1 km and 5 km resolutions are provided for modelling at coarser scales. It is a globally-consistent product that uses all available point data from the World Soil Information Service (WoSIS) database (Batjes et al., 2024), also from ISRIC-World Soil Information, and covariates with global coverage. Political boundaries are nowhere visible, except where one or more covariates match these. ···

*irrelevant*

315    SoilGrids provides both predictions and their uncertainty, via quantile random forest  models. It closely follows the *GlobalSoilMap* specifications of properties and depth slices (Science Committee, 2015). It also predicts the derived property of SOC stocks from 0-30 cm, in $T\ ha^{-1}$, computed from SOC concentration and bulk density. We chose to evaluate this layer, in order to compare it with the FAO's Global Soil Organic Carbon Map (GSOCmap) project (FAO, 2018).

We selected a transnational study area with corners (-109.99, 27.86) E and (-100.03, 35.64) N. This covers most of Chihuahua
320    and Coahuila and part of Sonora States (MX) and portions of Texas and New Mexico States (USA). Figure 14 shows this area,

*[handwritten annotation:] • Are all segmentations the "same"? • How can we better "see" the spatial structure of clustering (segments)? → All clusters/segments w/ color, no outlines, contour soil property over   overlay...?*

[Figure]

**Figure 13.** Generalised clusters of the segmentation of Figure 9, based on slicing the clustering dendrogram shown in Figure 12 for seven general clusters.

with the SOC stocks over the 0–30 cm depth slice. The higher stocks are in mountains and wetlands along the Rio Grande, the lower in high deserts.

Individual $2 \times 2°$ tiles of the 250 m product were downloaded in the GeoTIFF format from the interactive SoilGrids site (ISRIC-World Soil Information, 2024b), imported into R with the terra package, mosaicked, projected from the original geographic coordinates to a local Albers Equal Area projection, and trimmed to 3 270 x 3 610 6.25 $km^2$ pixels, covering 737 793.8 $km^2$. The global map of the 1 km product was downloaded in the GeoTIFF format from the ISRIC WebDAV repository (ISRIC-World Soil Information, 2024a), projected from the original Homolosine coordinate reference system to the same local Albers Equal Area projection, and trimmed to 900 x 900 1 $km^2$ pixels, covering 810 000 $km^2$.

Predicted SOC stocks per pixel ranged from 0 to 83, median 28 T $ha^{-1}$ for the 250 m product, and 7 to 76, median 29 28 T $ha^{-1}$ for the 1 km product, showing the smoothing effect of upscaling.

**Figure 14.** SoilGrids v2.0; SOC stock 0-30 cm, T ha$^{-1}$.

**4.1 Aggregation**

We applied the `supercells` algorithm to the SOC stocks 250 m resolution layer. To limit processing time and memory requirements, we selected a small test area of 80 x 80 km, i.e., 640 000 ha, centred on (-105 E, 32 N) at the Texas (N) / New Mexico (S) border, near Dell City NM (Figure 15). The centre pivot irrigated fields at the centre-left are $\approx 800 \times 800$ m and should thus be resolvable on the SoilGrids map. This area includes a wide range of the SOC stocks (Figure 16 left), with high values in the Guadalupe Mountains to the east and very low values in the salt flats in the centre of the area.

After some experimentation, a medium value (0.5) for *compactness* was selected. We did not set a minimum mappable area *mineara*, rather a number of proposed supercells $k$. A choice of $\approx 400$ supercells corresponds to an average area of 1 600 ha, corresponding to 1 cm$^2$ on a 1:400 000 printed map. This is much larger than the area of single centre-pivot irrigated fields, so we did not expect these to be individually resolved.

[Figure]

[Figure]

**Figure 15.** Test area for aggregation, centred on (-105 E, 32 N). Source: © Google Earth

Figure 16 (right) shows the computed supercells. Median size of the 412 supercells was 1 388 ha, ranging from 431 to 5 462 ha, with a strongly right-skewed distribution. This aggregation clearly groups the pixels with similar SOC concentrations. However, the shapes do not seem to correspond to natural landscape boundaries. We attempted other combinations of compactness and supercell numbers, with poorer results.

345     The quality of the aggregation can be measured by the standard deviation of the property within each supercell (Figure 17). These ranged from 0.34 to 6.08, median 1.18 T ha$^{-1}$, with corresponding coefficients of variation from 1.36 to 26.61, median 4.39%. The highest heterogeneity was in the pivot irrigation area, where the minimum supercell size forced pixels with a wide range of values together.

$\rightarrow$ *could This be an artifact of spectral covariates used in by SoilGrids?*

**4.2 Segmentation**

350     Segmentation was applied to the SOC stock map of the full study area, for both resolution SoilGrids DSM products. Since `gpat_gridhis` requires class maps, SOC stocks were classified in 19 (250 m) and 18 (1 km) equal intervals of 4 T ha$^{-1}$, with from 31 to 1'956 813 (250 m) and 14 to 128 549 (1 km) pixels per class. The minimum grid resolution for the 250 m product is here $2.5 \times 2.5$ km. The map was segmented at this resolution, and also four coarser resolutions: $5 \times 5$ km, $10 \times 10$ km,

[Figure]

[Figure]

**Figure 16.** SoilGrids v2.0 SOC stock (left) and its aggregation into supercells (right); SOC stock 0-30 cm, T ha$^{-1}$

[Figure]

**Figure 17.** Standard deviation within supercells; SOC stock 0-30 cm, T ha$^{-1}$

[Figure]

[Figure]

*I "nee" & no meaningful generalization of the landscape*

[Figure]

**Figure 18.** Segmentation of 250 m resolution SoilGrids map (part) at 1:1M nominal resolutions *(SOC)*

*\* would one expect meaningful generalization of the soil landscape based only on SOC stock ... I doubt it.*

20 × 20 km, and 40 × 40 km, corresponding to map scales 1:1M, 1:2M, 1:4M, 1:8M, and 1:16M, respectively. These produced

355   7 600, 1 905, 491, 127, and 35 segments from the 250 m resolution map, respectively. Figure 18 shows the segmentation at the finest scale. The level of detail is apparent, but many segments seem to be of a single class, with no internal pattern. Broader landscape patterns are obscured by this level of detail.

From the 1 km resolution SoilGrids map the three coarsest resolutions resulted in 669, 165, and 43 segments. Figure 19 shows these three segmentations. As resolution decreases, broader landscape patterns are increasingly aparent. All segmentations

360   seem useful at their respective design scales. *] Explain*

For the 250 m SoilGrids segmentation, median standard deviation increased from 2.35, 3.08, 3.94, 4.62, to 6.15 T ha$^{-1}$, while the median normalized Shannon entropy increased from 0.311, 0.369, 0.433, 0.472, to 0.580, for the 1:1M, 1:2M, 1:4M, 1:8M, and 1:16M scales, respectively. Entropy and standard deviation increase with segment size, as expected. The comparable values for the 1 km SoilGrids segmentation are median standard deviation 3.37, 4.29, and 6.19 T ha$^{-1}$, and median normalized

[Figure]

**Figure 19.** Segmentation of 1 km resolution SoilGrids map (part) at (left to right) 1:4M, 1:8M, and 1:16M nominal resolutions

[Figure]

**Figure 20.** Normalized Shannon Entropy of segments of the SoilGrids v.2 1 km map (part) at 1:16M nominal resolution.

365    Shannon entropy 0.417, 0.481 and 0.584 for the 1:4M, 1:8M, and 1:16M scales, respectively, similar to those from the 250 m segmentation.

Figure 20 shows the entropy for each segment of the 1:16M nominal resolution map from the 250 m product. This is a measure of the internal class homogeneity of each segment, although not the spatial pattern of the classes. The highest entropies are found in the segments with mixed high and low terrain, shown as contrasting purple and light blue colours.

370    **4.3   Clustering**

Figure 21 (left) shows the 39 segments signatures from the 1 km product, using motifel size 40 cells, and Figure 21 (right) shows the assignment to seven generalised clusters. Figure 22 shows the dendrogram for the clustering of the 39 segment signatures.

[Figure]

**Figure 21.** Left: Segmentation of SoilGrids v2.0 1 km SOC stocks (T ha$^{-1}$), motifel size 40 cells; Right: Assignment of segments to seven generalised clusters

[Figure]

**Figure 22.** Dendrogram of segment signatures, SoilGrids v2.0 1 km, motifel size 40 cells.

[Figure]

[Figure]

**Figure 23.** Jensen-Shannon divergence from Segment 1

The co-occurrence pattern of classes is similar within each general cluster. The clusters should group similar soil landscapes,
375  at least with respect to the SOC concentration. For example, cluster 1 groups mountainous terrain with high SOC interspersed
with basins with medium SOC in an intricate pattern.

Figure 23 shows the Jensen-Shannon divergence with the first segment, which necessarily has no divergence. These range
from 0.14 (segment 30, in the same cluster 1 as the target segment, although on a different first branch at height 0.45) to
0.84 (segment 4, in widely-separated cluster 3, different at branch height 1.45). This can be used to find the soil patterns that
380  are most similar to any segment, independently of cluster membership. The distance does not directly correspond to cluster
distance in the dendrogram when linkages other than single are used, as in this case, Ward's D2.

**4.4  Evaluation**

Aggregation was able to form compact groups of pixels with similar SOC stocks, which could be useful for, e.g., stratified
sampling. However the polygons did not seem to correspond well with landscape units. Segmentation was more successful.
385  At several increasingly-general scales it grouped distinctive patterns of SOC stocks, corresponding to large landscape units.
Clustering was then able to identify general groups of landscape units, and the Jensen-Shannon divergence identified the
segments most similar to a selected segment.

[Figure]

**5 Case study 3 – SOLUS100 (USA)**

The third case study is intermediate to the first two. Like the BIS-4D study it is of one country and with training points from one source, but (1) it covers a much wider area and so can't use covariates that are only available for part of the area, and (2) the product is based on numerous traditional soil surveys of varying age and quality control which can be used to some extent for evaluation.

SOLUS100 ("Soil Landscapes of the United States 100-meter") is a recent DSM product from the USDA-NRCS (Nauman et al., 2024). This contains predicted values, high and low estimates, and prediction intervals for soil properties at the *Global-SoilMap* standard depths, at 100 m horizontal resolution (i.e., 1 ha pixels) over the entire conterminous United States (CONUS). The maps are available in GeoTIFF format (Nauman, 2024). These can be compared to the Gridded Soil Survey Geographic Database (gSSURGO) digital product from the NRCS (NRCS Soils, 2022). This was created by digitising the polygons from traditional soil-landscape survey, with its linked relational database of polygons, map units, components, horizons, and soil properties. Thus aggregation and segmentation can be compared to a product based on expert judgement and field-based soil survey, although gSSURGO is also quite heterogeneous in the age and quality of the soil surveys on which it is based, and so must be used with caution as a ground truth.

We selected a 570 km² test area in Wayne County NY, mapped in 1978 on an unrectified airphoto base (Higgins, 1978), and later digitised by the NRCS and incorporated into gSSURGO. This area has a distinctive pattern of NNW-SSE orientated drumlins of various sizes and shapes, and inter-drumlin depressions. Some of these developed into peatlands, with drained areas used for agriculture and undrained areas used as wildlife reserves. The genesis of this soil landscape has been studied for more than a century (Menzies et al., 2016).

Figure 24 shows the predicted surface layer clay concentration for the original soil survey, as compiled in gSSURGO, and for SOLUS. Notice the different legend scales, otherwise the SOLUS map would not clearly show its pattern, since SOLUS predicts a narrower range of concentrations, as is typical of DSM products. It is obvious by visual inspection that SOLUS misses much of the fine pattern, and especially that it does not identify most of the organic soils with very low clay concentrations (dark blue on the gSSURGO map).

**5.1 Aggregation**

We aggregated the SOLUS map of surface layer clay concentration with the `supercells` algorithm. We set the minimum area parameter `minarea` to be comparable to MLD at original design scale. The source map in this area was at 1:24k design scale, so the MLD was set to 2.304 ha. The Optimal Legible Delineation (OLD) is 4 x MLD (Forbes et al., 1982), so here 9.216 ha, corresponding to nine SOLUS cells. Aggregation complexity is controlled by the number of supercells. This should be comparable to the number of gSSURGO polygons in this study area. In this way we can evaluate how well the DSM can match the traditional soil survey. In this area there are 14,949 gSSURGO polygons, with a median area of 2.43 ha, corresponding to 2 to 3 cells. The mean is area 5.30 ha, because of some large polygons, mainly organic soils, i.e. Histosols in US Soil Taxonomy (Soil Survey Staff, 1999).

**Figure 24.** Clay concentration % of the 0-5 cm layer, gSSURGO (left), SOLUS 100 m (right). Coördinate Reference System is an Albers Equal Area for CONUS.

We aggregated with a range of compactness values from 0.2 to 2. Because of the long linear shape of the drumlins, we expected that the lower compactness would best match the landscape. Indeed, this parameter value produced the map with the least rounded features, but their orientation did not match the landscape pattern (Figure 25). From this we conclude that SOLUS in no way represents the actual soil pattern. This same result was obtained with other layers of clay concentration, and

425 with several other soil properties.

[revised manuscript text omitted]

---

## Author Response (AR1)

General reponse to reviewers, egusphere-2025-1896

We thank the three reviewers for their close reading and useful suggestions. These will, we hope, substantially improve the manuscript and increase its usefulness.

Here we list the main changes proposed for a revised manuscript, synthetizing the most substantial of the reviewer's comments.

- 1. Discussion of how these methods are used in DSM evaluation, and their relation to point-based evaluation statistics; advice for their use.
- 2. Improved and added figures with more thorough explanations of what can be seen in them. Avoided colour schemes not suitable for colour blindness.
- 3. Replacing much of the in-text numerical results with tables, allowing easier comparison, with the main points brought out in linked text.
- 4. Recomputing in order to produce revised figures. This led to an improved version of \S3.2.3 "Multivariate segmentation with selected properties and depth slices" which is the example in \S3.3 "Clustering".
- 5. Better comparison of results from segmenting two resolutions of SoilGrids v2.0

\_\_\_\_\_

Answers to RC1: 'Comment on egusphere-2025-1896', Anonymous Referee #1, 25 Jun 2025 (Citation: https://doi.org/10.5194/egusphere-2025-1896-RC1)

We thank the reviewer for these corrections and thought-provoking larger questions about the focus of the paper.

- 1. It's nice to read a research highlighting the importance of evaluating the digital soil mapping (DSM) products from quantifying the spatial pattern of the predicted soil properties, besides the normal evaluation way based on sample-level error statistics, while many previous research also qualitatively discussed the reasonability of spatial pattern of DSM results.
- >> Reply: Thank you, that was our intention.
- 2. While the title "let the map speak for itself" is very attractive, I feel it would be more precise to say current research "let the map speak for itself under regulation of specific summarization ways", that is, aggregation and segmentation (then with hierarchical clustering). Also note the behavior (by design) of specific algorithm might introduce some unwanted effect or fail to reveal spatial patterns which might exist within the DSM results,

such as some algorithms might incline to identify round instead of long linear shape. Please consider to present or discuss it.

>> Reply: It is certainly correct that the analysts (in this case, the authors) do have to make decisions when applying the algorithms. Indeed, the selection of a "shape" parameter for aggregation is key, as is pointed out in the paper. Similarly, the motifel size for segmentation, and clustering algorithm choice. So how to modify the title, without making it too long? We propose changing the title to "Helping the map..."

So: "Representing soil landscapes from digital soil mapping products — helping the map to speak for itself"

- 3. I'm very interested in an in-depth discussion on comparison (as well as how to combine) between the aggregation way and the segmentation way tested in current research with three cases of different scales. Current manuscript has a good start and could strengthen this point.
- >> Reply: Indeed, we did not go into detail on how to use the two approaches: which to use when, and how they might be combined into a systematic evaluation.

We propose to add this to the Discussion section, with a topic sentence "So, how should aggregation and segmentation be used in an overall evaluation of a DSM product?"

Some of the proposed text: "The common use of point evaluation statistics by cross-validation or repeated data splitting is still important, as long as the representativeness in both geographic and feature space is clear to the map user... But as explained in the Introduction, these do not account for spatial patterns... An obvious evaluation of aggregation and segmentation can be the expert opinion of the soil geographer familiar with the mapped area... The starting point in any evaluation is the intended use(s) of the map. Then its fitness for use can be assessed according to the requirements to support those uses.... Pattern-based evaluation is indicated if be map be used to represent soil geography, for example, to help map users assess the relation of soils with the landscape. It is also indicated if the map user will need to identify landscape components, for example for ecological zoning of a protected area. The degree of internal heterogeneity as revealed by the segmentation can be used to assess connectivity, for example in catchment hydrological models.... One application where segmentation analysis must be used is identifying areas similar in their internal spatial pattern to a known area where the pattern has been characterized."

We intend to expand this text, but from here the outline of the argument should be clear.

**4. Other minor comments:**

Lines 20-22: As a general definition on digital soil mapping, it is not accurate. Note that some of DSM methods are not by fitting (geo)statistical or ML.

>> Reply: Agreed. Notably, the "similarity" DSM methods pioneered by A-Xing Zhu. We propose to change this to "by fitting geostatistical, machine-learning, or similarity-based models..." and add the reference Zhu, A.-X. and Turner, M.: How is the Third Law of Geography different?, Annals of GIS, 28, 57-67, https://doi.org/10.1080/19475683.2022.2026467, 2022 as an introduction to the similarity-based concepts. This paper contains references to several studies using this approach.

Section 1: Paragraphs since Line 77 are actually to present the work of this study. It would be clear to say that here, like "this study ...", instead of say it until the last paragraph of Section Introduction.

>> Reply: Correct, it's not obvious at this point that this is what we will do. We propose to change L77 to "In this study, we examine two ways to assess the success of DSM in reproducing a soil landscape. The first is to aggregate..."

Then we propose to change the next paragraph's topic sentence to "The second way is applied at coarser scales, where the homogeneity of properties within some larger area may not be possible or even desirable."

In this way it's clear that both are our methods and the distinction between them.

Line 205: "All 54 maps (7 properties, 6 layers)" — ? 9 properties?

>> Reply: Yes, that's correct. This will be corrected to "9". Thanks for noticing this. "All 54 maps (nine properties, each with six layers)..."

Lines 265-266, "we selected key properties at key depths: pH, clay, silt, SOM 0-5 cm, clay, bulk density 15-30 cm, CEC 30-60 cm, sand, SOM 100-200" — to be clear on the depth of each soil property.

>> Reply: Yes this was a bit ambiguous. The proposed revised text is "... hence we selected key properties at key depths: (1) pH, clay, silt, SOM at  $0-5\sim$ cm; (2) clay and bulk density at  $15-30\sim$ cm; (3) CEC at  $30-60\sim$ cm; and (4) sand and SOM at  $100-200\sim$ cm."

We also propose to add some justification for choices that may appear unusual: "The reason for using SOM of the deepest layer is to distinguish thick peats, and for using sand of that same layer is to distinguish thick dune sands."

\_\_\_\_\_

Reply to RC2: 'Comment on egusphere-2025-1896', Anonymous Referee

**2, 30 Jun 2025, Citation: https://doi.org/10.5194/egusphere-2025-1896-RC2**

We greatly appreciate the detail with which the reviewer examined the paper.

- 1. This paper introduces and applies a pattern-based evaluation approach for Digital Soil Mapping (DSM) products, built around the concept of letting "the map speak for itself." Through three distinct case studies—BIS—4D in the Netherlands, the global SoilGrids v2.0 product, and SOLUS100 in the USA—the authors explore how aggregation, segmentation, and clustering methods can be used to evaluate whether DSM products reflect meaningful soil landscape patterns. This approach complements traditional point—based accuracy assessments by focusing instead on spatial coherence and landscape—level pattern recognition.
- >> Reply: Thank you, that is a good summary of the paper.
- 2. One of the strengths of this paper is that it goes beyond traditional point-based accuracy metrics and puts emphasis on evaluating the spatial realism of DSM outputs. This aspect is often neglected in many DSM studies, where maps are validated statistically, but not checked visually or structurally to see if they actually reflect real soil landscape patterns.
- >> Reply: Indeed this is the aspect we are trying to address. This paper is the second effort, following the paper (cited) which uses some (geo)statistical methods to characterize the patterns in maps. Here we go further to examine the patterns themselves, as extracted by the two presented algorithms.
- 3. The case studies are carefully chosen to illustrate the strengths and limitations of the methods under different mapping conditions: one that performs well (BIS-4D), one global product with mixed success (SoilGrids), and one that fails to capture the landscape structure (SOLUS). This comparative structure strengthens the paper and offers valuable insights for practitioners working with DSM.
- >> Reply: Thank you. However, the products could have performed differently, with different levels of "success", in other case studies. In particular, recently deglaciated areas (chosen for the SOLUS example) could pose a problem for any DSM product. We propose to add a sentence to the end of \S5.4 to emphasize this:

"This is not meant to be a condemnation of SOLUS as a useful product overall. All DSM models trained over a wide area can have difficulty when applied to a local area with idiosyncratic soil—landscape relations which are not reflected in the covariates available over the entire training area. This is a general "global model applied to locally—idiosyncratic landscapes" issue. In other areas of the USA SOLUS appears (by visual inspection) to well—represent the soil landscape."

- 4. That said, the paper can be quite dense and difficult to follow in places due to the high level of technical detail. Long descriptive passages filled with numeric results might be better supported by well-structured summary tables. For example, in the Netherlands case study, a table showing different properties, segmentation scales, and the resulting number or size of segments would allow readers to quickly scan and compare outcomes across scenarios. Presenting complex results visually or in tabular form would improve readability and accessibility.
- >> Reply: Correct. Indeed, with tables we can show more results for each experiment, which are then easier to interpret. We propose to do this throughout and use the text for commentary, not detailed results.
- 5. Figures are generally effective, Figure 11 (left), for instance, is particularly compelling, showing detailed patch delineation that appears well aligned with conventional maps. This kind of visual confirmation is rarely provided in DSM studies, yet it plays a crucial role in demonstrating the practical utility of such products. The paper succeeds in making a strong case for including spatial pattern analysis in DSM evaluation workflows.
- >> Reply: Thank you. We have modified the figure captions according to your suggestions (below) and those of other reviewers.

We agree that visualization is crucial, and this is not our specialty, although we have tried to learn from good examples in other papers and reports, as well as sound cartographic principles.

- 6. In summary, the study is a valuable contribution to the DSM literature, highlighting a needed shift toward landscape—aware evaluation. It will be especially useful for practitioners seeking to validate their maps beyond traditional metrics and foster broader acceptance of DSM by bridging the gap between digital outputs and expert expectations from conventional mapping.
- >> Reply: Thank you. We hope others will expand and improve this line of work.
- 7. I would recommend this paper for publication in EGU Sphere, pending minor adjustments. Please find my detailed comments below:
- >> Reply: Thank you for the positive recommendation. We have appended the comments from the reviewer's PDF here, and answer them.

**Introduction**

General terminology: Readers outside soil geography may be unfamiliar with specialized terms such as supercells, consociation, and SCORPAN. Please provide a brief parenthetical definition (or a short glossary) on first mention of each term.

>> Reply: Indeed we wrote this for researchers within the

discipline, but the paper would be more generally useful if we briefly define the terms. We didn't want to "talk down" to our direct colleagues, but some small explanation won't be out of line.

Proposed revisions, at the first (non-Abstract) mention of the terms:

- (1) Supercells: "...aggregate individual predictions from pixels into more or less homogeneous contiguous groups of pixels referred to \emph{supercells}..."
- (2) Consociation (and related): "Depending on the scale of the analysis and the inherent scale of the soil landscape, we may expect to see homogeneity at the level of map delineations containing dominantly one soil type within defined limits at a detailed categorical level (e.g., soil series, the lowest level of Soil Taxonomy); this is called a \emph{consociation} in the US soil survey \citep{soil\_survey\_division\_staff\_soil\_2017}, At a coarser scale we may expect a regular pattern of contrasting soil types forming a soil \emph{association}, or a fine-scale pattern of contrasting soils forming a soil \emph{complex}. These terms are also from the US soil survey. All three terms are well-explained, with examples, by Van Wambeke & Forbes(1986)."
- (3) SCORPAN: "DSM replaces the conceptual model with correlative relations with digital coverages meant to represent, at least in part, one or more of the seven predictive ``SCORPAN'' factors of McBratney et al. (2003)}. In this widely-cited paper they briefly describe as these factors as: '\textbf{s}: \emph{soil}, other properties of the soil at a point; \textbf{c}: \emph{climate}, climatic properties of the environment at a point; \textbf{o}: \emph{organisms}, vegetation or fauna or human activity; \textbf{r}: \emph{topography}, landscape attributes; \textbf{p}: \emph{parent material}, lithology; \textbf{a}: \emph{age}, the time factor; \textbf{n}: \emph{space}, spatial position. Note that these are correlative, not necessarily causitive, and are used to build a predictive model for mapping, not (at first) to understand pedogenesis. Thus in DSM there is no longer an explicit relation with the soil landscape, but it is hoped that the implicit correlative relations, based on representative covariates, can find these."

Line 40: The phrase "catena of Milne" could confuse readers who do not know the historical context. Add the publication year to anchor the historical reference. Consider wording such as: "...the classic example is the catena, as defined by Milne (1935), meaning 'a sequence of distinct but pedogenetically related soils consistently located on specific slope facets...' (Borden et al., 2020)." This makes it explicit that "Milne" refers to Geoffrey Milne's original definition of a catena.

>> Reply: Thank you, we always enjoy going back to the original concept, although the Borden et al. paper is more accessible and modern. Proposed wording: "The classic example is the catena as

defined by Milne (1935) as: ``a sequence of distinct but pedogenetically-related soils that are consistently located on specific slope facets, giving recurrent topographically-associated soil pattern'' (Borden et al.\, 2020)".,

Line 50: "At increasingly detailed scales and with increasingly fine distinctions in the definition of soil bodies, increasingly finer patterns are revealed" This sentence is conceptually clear but stylistically repetitive. The repeated use of "increasingly" and "fine" makes it feel redundant and slightly difficult to follow. I suggest rephrasing it for clarity and flow.

>> Reply: We propose to rephrase as: "With increasingly detailed cartographic scales and categorical definitions of soil types increasingly finer patterns can be shown. Conversely at coarser scales and broader categories patterns are necessarily more general." This is the message from Hole and Fridland.

Line 97: The objectives paragraph states that the study will derive soil—landscape units via aggregation, segmentation, and clustering, and that these units can be used for routine DSM evaluation. However, it is unclear how the derived patterns will be incorporated into the evaluation itself (e.g., spatial correspondence to legacy polygon maps, pattern—matching indices, error aggregation by unit). A brief statement of the specific pattern—based metrics or comparison framework you intend to use would help readers see exactly how the new units feed into DSM evaluation.

>> Reply: The last sentence of this paragraph states "Finally, we discuss how these methods can be used in routine evaluation of DSM products." We propose to add a link to the "Discussion" section, so the reader can click through to it. Then in that section we propose to add a final paragraph explaining how the information from aggregation and segmentation might be used in an overall evaluation. First proposed sentence "So, how should aggregation and segmentation be used in an overall evaluation of a DSM product?" We propose to add (1) still can use point-based statistics but with caution of their geographic/feature space distribution; (2) expert opinion of the soil geographer familiar with the mapped area; (3) all evaluations must refer to the intended use(s) of the map; (4) identifying areas similar in their internal spatial pattern to a known area where the pattern has been characterized.

Materials and Methods:

Line 105: Typo - "Aggregnation" should be corrected to aggregation.

>> Reply: How did we miss this? Thank you, will correct.

Line 109: Typo — Remove the duplicate "in" in "defined in in multivariate space."

>> Reply: Same

Line 109: Remove the stray period after the Lin (1991) citation.

>> Reply: Same.

Line 125: The sentence "Unlike supercells, segments must have rectilinear borders" could benefit from a brief explanation of the cartographic rationale behind this constraint. Why is this a requirement?

>> Reply: This was not clearly expresed; supercells also have rectilinear borders when zoomed into the pixel level. The point here is that segments must be of a certain minimum block size, which is constrained by the GeoPAT algorithm to be a square. Contiguous square blocks can then be aggregated. We propose to change this sentence to express this: "Patterns are computed within square blocks of at least 10 x 10 pixels; larger squre blocks can be specified by the analyst. This minimum block size and shape is required by the GeoPAT algorithm to be a square. Contiguous square blocks can then be aggregated into rectilinear segments.

Figure 2: Please spell out the abbreviations "gpat\_gridhis" and "gpat\_gridts" in the caption (and first mention in the text), so readers immediately understand what each tool does within the workflow.

>> Reply: We propose to add plain-language descriptions in quotes in the caption and text.

Line 143: For smoother readability, rephrase to: "Two important thresholds for joining grid cells into segments are:"

>> Reply: Agreed.

Line 146: "Spatial" does not need to be capitalized here.

>> Reply: Agreed.

Lines 175 - 177: Avoid using the vague term "better." Clarify what "better" means in this context, does it refer to lower inhomogeneity, higher isolation, or both? Clearly define the direction of desired improvement for each metric to avoid ambiguity.

>> Reply: Agreed. Proposed text: "Inhomogeneity measures the degree of mutual dissimilarity between a segment's motifels, on a [0 ... 1] scale, where smaller values correspond to more homogeneous and less internally diverse segments. Isolation is the average dissimilarity between a segment and its immediate neighbours, on a [0 ... 1] scale, where larger values correspond to segments that are more isolated from their neighbours. The most successful segmentation would have the smallest inhomogeneity and largest isolation."

Figure 3: Again, ensure that abbreviations like "gpat\_polygons" and "gpat\_distmtx" are spelled out in the caption, especially for readers unfamiliar with the GeoPAT suite.

>> Reply: Yes, please see reply to Figure 2 comments.

Consistency issue: You shift between past and present tense ("Here we used..." vs. "Here we use..."). Choose either past or present tense and apply it consistently throughout the Methods section.

>> Reply: We have tried to make this consistent.

Case studies:

Case study 1 - BIS-4D (Netherlands)

Line 205: 54 maps (7 properties, 6 layers): fine, though "depth layers" may be clearer than "layers".

>> Reply: Actually, it's 9 x 6 = 54 (our mistake, corrected), adding "depth" will make that clearer.

Line 211: minarea was to 1 600 25 m x 25 m pixels"  $\rightarrow$  typo  $\rightarrow$  should be "minarea was set to 1,600 pixels (25 m × 25 m).

>> Reply: Agreed, will be corrected. Proposed text: "Thus the \emph{minarea} parameter was set to 1,600 pixels, each of 25~m~x~25~m."

Line 221: classification of soil property maps for example: pH (0.1), clay (5%), etc., are arbitrary to the reader and not clear enough how you choose these values. Provide a brief justification (measurement precision, agronomic relevance, histogram breakpoints).

>> Reply: These would be chosen according to the evaluator's criteria. We propose to add "..., to illustrate this method we classified the soil property maps..." to the topic sentence, to show the arbitrary choice, and then to add some explanation: "In practice, the map evaluator would select class limits to correspond to the desired precision and thresholds for interpretations or models. The class widths can not be finer than the precision of the corresponding laboratory analyses."

We propose to add the example of Cornell University's liming recommendations for field crops, where the precision is 0.1 pH as used in our paper. Reference: Ketterings, Q., & Workman, K. (2023). Lime Guidelines for Field Crops in New York (p. 20). http://nmsp.cals.cornell.edu/publications/extension/LimeDoc2023.pdf

Line 264: isn't it 54 layers?

>> Reply: Yes, will be corrected.

Line 265: It is not clear which depth for which soil properties has been used, rewrite the whole sentence: for example, pH, clay, silt and SOM for the depth of 0-5 cm, clay and bulk density for the depth of 15-30 cm and continue. Otherwise, the reader would be confused.

>> Reply: Yes this was not clear. Will be rewritten as: "... hence we selected key properties at key depths: (1) pH, clay, silt, SOM at  $0-5\sim$ cm; (2) clay and bulk density at  $15-30\sim$ cm; (3) CEC at  $30-60\sim$ cm; and (4) sand and SOM at  $100-200\sim$ cm." We propose to add some justification for the last set: "The reason for using SOM of the deepest layer is to distinguish thick peats, and for using sand of that same layer is to distinguish thick dune sands."

Line 281: As you mentioned it, segmentation is greatly affected by two thresholds, it doesn't seem that default thresholds and liberal threshold would produce comparable maps. As I don't see the segments capture the distinct pattern in figure 11 (right) with more liberal thresholds.

>> Reply: We propose to update this section with a more comprehensive comparison, i.e., the two thresholds for four scales for a representative property (clay, all depths together) and present the results in a table with some commentary. Figure 11 will remain, it well-illustrates the effect of changing the thresholds.

Section 3.3 Clustering: First paragraph: It is not clear how to follow cluster numbers? And how you link each cluster to the soil landscape components? But also state how the number of clusters is selected for this case.

>> Reply: A justification for generalizing to seven clusters will be given, as well as a more thorough discussion of their landscape relations. That link is by comparison with the known soil landscape as explained in the introduction this case study.

Line 305: "due to the extremely high quality", it would be nice to add the quality of source data for better comparison.

>> Reply: This is explained in the Helfenstein source. We proposed to add "... as explained by Helfenstein et al. (2024)". The reader can find the detailed justification of "high quality" there.

In general, it would strengthen the paper to include basic descriptive statistics and accuracy/error metrics for the DSM products or baseline maps used in each case study. For example, if the maps are described as having "good accuracy," providing supporting metrics—such as RMSE, R², or standard deviation for selected properties (e.g., pH)—would give readers a clearer understanding of the underlying data quality. These statistics could be included in the Supplementary Materials if space is a concern, but they would offer helpful context when interpreting the segmentation and aggregation results.

>> Reply: This is a good suggestion. We think one or two sentences per case will give a sufficient picture of the product's quality.

For BIS-4D (Case 1) we propose to refer to the accuracy tables (7, 8) in Helfenstein et al., and to highlight several properties, e.g.

"For example, the 10-fold cross-validation average for all predictions of pH had a median ME of -0.023 pH, median RMSE of 0.72 pH, and a median MEC of 0.72. For clay these accuracy statistics are  $0.42\$ ,  $7.7\$ , and 0.78, respectively."

For SoilGrids SOC stock (Case 2) we propose to add: "Poggio et al., 2021a (Table 4) shows that SOC concentration had a median global cross-validation RMSE of 3.97\% and bulk density of 0.19, averaged over the three layers which contribute to SOC stock estimates."

For SOLUS (Case 3) we propose to refer to supplementary table S1 in Naumen et al. and report key cross-validation statistics for the property examined in Case 3, i.e. clay 0-5 cm. "Accuracy statistics are not available for this area, however, for this property over the entire CONUS (Naumen et al, 2024, Table S1) reports spatial cross-validation statistics of 6.481\% RMSE, -0.003\% ME, and 0.672 \$ \mathrm{R}^2\$, based on all 484~258 observations.

When compared to only the  $37\sim992$  observations that were analyzed in the NRCS Soil Characterization Laboratory these results were substantially worse:  $8.382\$  RMSE,  $0.011\$  ME, and  $0.544\$  \mathrm{R}^2\$"

Case study 2 - SoilGrids v2.0 (Global)

Line 338: mineara" → typo → should be minarea

>> Reply: How did we miss this? Will be corrected.

Line 356: "many segments seem to be of a single class" → vague. Maybe clarify in this way: "...many segments contain only one SOC class, offering little internal variability."

>> Reply: We propose to revise as "... but many segments contain only one SOC class, and thus have no internal pattern."

Figure 17: the highest standard deviation appears to be approximately 6.1, but in the text (Line 346), it is stated that the standard deviation ranges from 0.34 to 6.08. Please double-check this value, there may be a small discrepancy between the figure and the text.

>> Reply: The figure was created with rounded values for readability. We propose to add this information to the caption, thereby removing the discrepancy.

Section 4.4: While Section 4.4 offers a qualitative summary of the results from aggregation, segmentation, and clustering, it remains unclear how these steps contribute to the evaluation of DSM products, which is a central stated goal of the paper. The discussion is primarily descriptive and does not explain whether the derived spatial units are being used to validate DSM predictions, identify mapping artifacts, support field design, or quantify model accuracy. To strengthen the study's impact, I recommend adding a

brief clarification of the intended evaluation framework— for example, whether it involves comparison to legacy soil maps, pattern—based quality metrics, or expert validation. This would help readers understand how these methods serve as tools for DSM assessment rather than just spatial analysis outputs.

>> Reply: The other reviewers have also raised this point. We propose to add to the Discussion (not here in \S4.4) how these methods can be used in DSM evaluation, and their relation to point-based evaluation statistics. We don't see how to "quantify mapping accuracy" but the discussion of identifying mapping artifacts and supporting field design can be included.

Case study 3 - SOLUS100 (USA)

Line 419: The mean is area 5.30 ha -> the mean area is

>> Reply: Will be corrected.

Line 424: The statement "From this we conclude that SOLUS in no way represents the actual soil pattern" may be too strong without formal pattern—matching metrics or more comprehensive property comparisons. Consider softening this claim or supporting it with a visual or statistical metric beyond visual inspection.

>> Reply: Correct, there is nothing formal here, especially since we don't know the "true" pattern — although in this example area the landscape has highly distinctive features. We propose to add the information "...their orientation did not match the generally NNW—SSE pattern of the drumlin field." — this is clear from the referenced figure.

The authors clearly demonstrate the challenges of applying the DSM approach of SOLUS100 in a glaciated landscape with fine-scale patterning. Despite methodical application of aggregation and segmentation, the results did not reflect landscape features accurately, suggesting a mismatch between the DSM product's input data and the local soil-forming processes.

>> Reply: We mentioned this in \S5.4: "This is likely because SOLUS lacks locally-important covariates to represent this recently glaciated soil landscape with its characteristic drumlins."

**Discussion:**

The authors acknowledge limitations in parameter selection, but what guidance can they offer for users who are not yet experts? This study suggests that appropriate parameter selection is crucial for success, otherwise, the methods may fail to produce meaningful results. For instance, the text states, "There is no objective way to adjust compactness and supercell number parameters." It would be helpful if the authors could recommend whether any metrics (e.g., internal variability, boundary length, or other summary statistics) might serve as semi-objective tools for parameter tuning, even as

part of a sensitivity analysis.

>> Reply: The idea of sensitivity analysis is good, but beyond what we have space for in this paper. We approach this for segmentation with the different choices of size and motifel in Cases 1 and 2. One recommendation for aggregation shape is to match the shape of prominant soil landscape features, e.g., the drumlins of Case Study 3.

We propose to add to the Discussion:

"An obvious question is how to parameterize the two approaches. In this paper we compared several choices of parameters in each case study on an \emph{at hoc} basis. It may be possible to systematize this with sensitivity analysis, to quantify the changes in results as parameters change. However, this was outside the scope of this paper."

The discussion refers to traditional mapping units such as polypedons, consociations, complexes, and associations. It would be helpful if the authors clarified whether their derived units are intended to correspond to these traditional concepts, or whether they serve a different function within the context of DSM evaluation. As the authors note, traditional surveys themselves can be inconsistent—particularly when comparing products like gSSURGO in the U.S. and the INEGI map in Mexico. This raises the question of which map should be treated as the baseline for comparison, especially when these traditional sources vary in quality and methodology.

>> Reply: This is quite tricky. All soil surveyors and most field scientists using soil maps (e.g., ecologists) soon recognize that some conventional maps are more reliable than others. So just matching a conventional map at the appropriate degree of generalization is not always appropriate. In the Case Study 3 the landscape and soil patterns are highly distinctive so that the original surveyors could hardly make mistakes — the only problem could be digitizing from the unrectified photo base used for the original survey to a correct topographic base for incorporation in SSURGO. We propose to discuss this in the new subsection of Discussion on how these methods can be used to complement point—based methods for DSM evaluation.

Some of the proposed text: "The common use of point evaluation statistics by cross-validation or repeated data splitting is still important, as long as the representativeness in both geographic and feature space is clear to the map user... But as explained in the Introduction, these do not account for spatial patterns... An obvious evaluation of aggregation and segmentation can be the expert opinion of the soil geographer familiar with the mapped area... The starting point in any evaluation is the intended use(s) of the map. Then its fitness for use can be assessed according to the requirements to support those uses.... Pattern-based evaluation is indicated if be map be used to represent soil geography, for

example, to help map users assess the relation of soils with the landscape. It is also indicated if the map user will need to identify landscape components, for example for ecological zoning of a protected area. The degree of internal heterogeneity as revealed by the segmentation can be used to assess connectivity, for example in catchment hydrological models... One application where segmentation analysis must be used is identifying areas similar in their internal spatial pattern to a known area where the pattern has been characterized."

We intend to expand this text, but from here the outline of the argument should be clear.

**Conclusion:**

The conclusion effectively summarizes the intent and broader value of the methods, particularly the shift toward spatial pattern-based evaluation of DSM products. However, the claim that the map is allowed to "speak for itself" might be reconsidered or clarified, since the process still depends heavily on analyst-defined parameters.

>> Reply: Yes, this is the point you make in the general comments. See response there.
* * *
Answer to RC3: 'Comment on egusphere-2025-1896', Dylan Beaudette, 30 Jun 2025, Citation: https://doi.org/10.5194/egusphere-2025-1896-RC3

We thank Dylan Beaudette for his thorough reading of the paper, many suggestions for improvement, and especially highlighting links to earlier soil geography concepts and current NRCS practice.

1. I'd like to thank the authors for submitting such a creative approach to investigating the reliability (even credibility) of spatial models for soil properties. For too long we have relied on point-based evaluation of these models, despite the fact that the geosciences have more than a century of reliable frameworks for explaining the geography of the Earth's surface form and function. A formalized, pattern-based evaluation of spatial models would immediately benefit both soil science and soil survey.

I think that this manuscript should be accepted after minor revisions.

Please see detailed comments below, and the attached PDF with hand-written annotations and notes. There are many, small comments, ideas, and suggestions that are only present in the annotated PDF.

Dylan Beaudette Research Soil Scientist USDA-NRCS >> Reply: Thank you for the encouragement, especially coming from an NRCS scientist intimately involved with the US soil survey, one product of which is our Case Study 3. And thank you for the detailed reading and commenting. You bring up important conceptual issues, as well as technical issues with the manuscript, particularly with the graphics. We have answered your comments here. Comments in the marked—up PDF that are not repeated here are added at the end of this text and answered there.

**## Overall Comments**

- 1. Spatial modeling of soil properties (I guess we have to call it DSM?) has been happening since before McBratney et al, 2003—might be nice to cite some earlier studies when setting a historical context. Examples that come to mind are the work of Hole, Arkley, Bishop, and others from 1960s 1990s.
- >> Reply: Hole (and Fridland) are mentioned at L50 in relation to pattern analysis, the focus of this paper. Indeed, Hole (with Campbell) published a book on soilscape analysis in 1978. This can be cited as an example of using landscape correlatives to identify distinct soil bodies. Good idea and we like to bring in history when possible. We propose to add a sentence to the end of the first paragraph:
- "DSM is a digital, semi-automated form of landscape analysis as used in traditional soil survey to identify distinct soils from environmental covariates (Hole & Campbell, 1985)."
- 2. I think that we all understand what "machine-learning" is supposed to mean, but the term is so widely used that the meaning has become muddied. What do the authors think about "statistical models" or "probabilistic models" when describing this kind of work?
- >> Reply: Good idea. "Statistical learning" in the sense of the famous Hastie & Tibshirani text (will be cited) is a better term, and will replace "machine-learning" at L21, and the term "machine-learning" can be removed at L315 where SoilGrids models are explained.
- 3. Along those lines, starting sentences with "DSM predicts..." is an awkward construction and makes it sounds like there are no humans involved.
- >> Reply: We propose to rephrase this as "Since predictions DSM are per pixel..." and "Digital Soil Mapping (DSM) products show predicted values of soil properties or classes ..." We think it's clear that DSM does not happen by itself (at least not yet...).
- 4. Since the multi-panel figures share a common extent, it should be possible to remove coordinate axes labels and make note of the BBOX in the caption. That would leave more room for larger subfigures. It is very difficult for me to read or see enough detail in individual panels.

- >> Reply: We prefer to retain the coordinate axes labels so that users can find these locations in other products (e.g., Google maps) if they want to examine them.
- 5. It seems like using soil organic carbon for an example investigation is problematic. We know that SOC varies tremendously within a single pedon (e.g. horizon topography, bioturbation) or over very short horizontal distances. The analysis of longer-range spatial connectivity or "regionalization" of SOC is further complicated by the relatively small range of expected values in arid regions. Could an alternative soil property such as pH, sand content, gravel content, or CEC have been used instead? Or, if the authors prefer to stick with SOC, please consider adding more discussion of the known patterns in SOC (orographic / bio-climatic / desert -> forest) and how well the predictive maps tracked such patterns.
- >> Reply: In an earlier draft of the paper we compared SoilGrids globally-consistent map to the FAO's Global Soil Organic Carbon patchwork (by country) map. As explained "We chose to evaluate this layer, in order to compare it with the FAO's Global Soil Organic Carbon Map (GSOCmap) project (FAO, 2018)"; however, evaluating that map made the paper too long and didn't contribute much to its message, so we removed it. Still, it's a good map to evaluate, see the comment below for a discussion of the spatial aspect raised here. The high interest in SOC and the many efforts to map it justify its choice. We propose to change the justification to: "We chose to evaluate this layer due to its intrinsic importance, as evidenced by the efforts of the FAO to produce a global map from national contributions in the Global Soil Organic Carbon Map (GSOCmap) project, see a portion of that map in Figure 28, below."

We propose to further explain that we selected SOC exactly because it is difficult. It is modelled by many groups and using many method. The results are used in many policy applications — how can the user evaluate these competing products? We propose the relation to the soil landscape.

- 6. The description of detailed soil survey within the US is, in general, quite accurate. However, It is important to note that most of the Soil and Plant Science Division staff have been working on updates since 2013. Any remaining "join issues" are still "work-to-be-completed" as of June 2025.
- >> Reply: This refers to L393-401 where the US soil survey inputs to SSURGO are described. We propose to add a sentence explaining the updating, as pointed out by the reviewer.

Proposed text: "NRCS has been working on updates to source maps as well as harmonizing map unit names and boundaries across different survey areas since 2013, although this work is not complete. These updates are then used in new versions of gSSURGO."

- 7. A short description and discussion of how the observations in this paper relate to the tension between global vs. local modeling. For example, is it realistic to expect a global or continental model to produce meaningful predictions below some minimum resolvable area? In other words, can the utility (and confidence) of these kind of models be improved by a well-specified minimum resolvable area?
- >> Reply: This is explained for SOLUS at L449, see below. We don't see how a general discussion of this point can be justified from this paper.
- 8. While not a perfect match, the "aggregation" and "segmentation" concepts here are closely related to similar operations in traditional soil survey. Aggregation over soil properties and landform elements gives rise to soil component concepts, and segmentation over spatial units gives rise to soil mapping units.
- >> Reply: This is good although as you say imperfect analogy. And, in traditional soil survey these are applied at different survey orders: aggregation at (semi-)detailed survey and segregation at reconaissance scales. The Introduction discusses the application of the two methods at different scales. We propose to add wording similar to this comment immediately after that discussion. It's good to make that link.
- 9. Please check all figures containing maps for units of measure, and labeling of color scales.
- >> Reply: Yes, that is covered in your detailed comments, see below.
- 10. Do not be timid about interpretation and critique of these data, both traditional soil survey and statistical prediction. From the point of view of a soil scientist and soil surveyor, much of the confidence that I place in our products is based on witnessing the internal gauntlet of scrutiny required before publication.
- >> Reply: We don't have any "inside information" on how any of the products were produced, except of course for SoilGrids, the workflow and quality control of which is documented in a referenced paper. Our aim in this paper is not to evaluate the products as such, but to show how they can be made to reveal spatial patterns, which can then be evaluated, at this point only by experts but later perhaps by semi—automated methods. So we don't think it's appropriate here to critique beyond what we can see about the success of the segmentation and aggregation. We already have statements on these for each case study.

**## Specific Comments**

1. Lines 30-45: it is important to note that there are as many, if not more, abrupt changes in soil properties which are controlled by process notoriously difficult to model: lithology, landform age, faulting, and so on. Assuming that soil variability should be modeled only as a continuum sets up statistical models for low

reliability and confidence.

>> Reply: Good point, it gives us a chance to cite one of our favourite papers on soil boundaries: Lagacherie, P., Andrieux, P., & Bouzigues, R. (1996). Fuzziness and uncertainty of soil boundaries: From reality to coding in GIS. In P. A. Burrough, A. U. Frank, & F. Salgé (Red.), Geographic objects with indeterminate boundaries (pp. 275–286). Taylor & Francis.

We propose add after L35 "... natural soil bodies according to those limits, which may be may be abrupt or smooth (Lagacherie et al., 1996), according to the spatial pattern of the soil-forming factors."

- 2. Lines 45-50: must we refer to "soil-forming factors and process" by ClORPT or SCORPAN? This very much feels like referring to a product by brand name vs. connotative description.
- >> Reply: Since Jenny and McBratney these are widely—used and understood. Another reviewer suggested a brief description of them, to help readers who are not so familiar with DSM. This we propose to do. Also, we will make it clear that the SCORPAN abbreviation is widely—used (for better or worse) and understood in the DSM community. Another reviewer asked for a brief definition of SCORPAN factors for the general reader, and we propose to add that.
- 3. Lines 55-60: the repeating nature (or not) at some target scale is often referred to as "mappabilty" by soil surveyors. It is a squishy definition for what (I think) the authors are trying to formalize here.
- >> Reply: Correct. However, we don't see how to work that concept into the flow of the text here.
- 4. Lines 65–70: it can be useful to separate the spatial precision (minimum resolvable area) from conceptual precision (MU kind  $\sim$  consociation, association, complex, undifferentiated group) when discussing spatial prediction via expert system or statistical model.
- >> Reply: Yes, the (mis-)match between cartographic and categorical generalization is a key issue in any mapping exercise. We think that this is satisfactorially explained around L85 where the mapping unit kinds are explained, and we state "Depending on the scale of the analysis and the inherent scale of the soil landscape...". We propose to change the ambiguous "scale" to "scale (i.e., for DSM, the horizontal resolution) ..."
- 5. Line 85: "homogeneity of the land cover pattern" is very similar to the soil survey "soil mapping unit" concept. It is worth drawing this parallel between imagery analysis, human vision, and traditional soil survey.
- >> Reply: We don't think this interesting parallel can be introduced

- here without disrupting the flow of the argument leading to our study.
- 6. Lines 150-165: it could be useful to link these ideas to "survey order" from the US soil survey program, and Soil Survey Manual.
- >> Reply: Good point, especially for USA readers. At the end of the paragraph (L161) we propose to add: "These concepts are comparable to concept of soil survey orders in the USA soil survey (Soil Survey Division Staf, 2017, Chapter 4) and the ``resolutions and extents for DSM'' of McBratney et al. (2003, Table 1)."
- 7. Line 219: can the authors please expand on this statement? How is "importance" evaluated?
- >> Reply: This is a feature of the Jensen-Shannon divergence algorithm. So we propose to clarify this as "...has a small contribution to of Jensen-Shannon divergence in that supercell."
- 8. Figure 7: these are too small to interpret, perhaps select 2 design scales to focus on?
- >> Reply: We wanted to show the effect of design scale at three levels. The solution we propose is to convert this from a 1  $\times$  3 to a 2  $\times$  2 format and add another coarser scale. We plan to ammend the description to discuss this coarsest scale as well.
- 9. Figure 8: the reader (this reader) will need more explanation / assistance with the interpretation of these concepts and what the maps mean.
- >> Reply: At L130 we briefly explained "Finally, the result is evaluated by its segmentation statistics, namely, inhomogeneity within the segment and isolation of the segment from its neighbours". Then in the description of the GeoPAT suite at L173ff. is a paragraph explaining these measures. So by L253 (reference to Figure 8) the meaning of the figure has been explained.
- 10. Lines 265-270: I suggest expanding on the concept/experience of grouping soils "holistically" vs. by-property.
- >> Reply: Yes, this paragraph started with what we did, not why we did it and what is the difference with the previous section. We propose to add a new topic sentence at the beginning of the paragraph:"Although BIS-4D predicts each property separately, the soil as a natural body is of course more than a stack of individual properties, and this is recognized by the concept of diagnostic horizons and properties in modern soil classification systems, and soil series in detailed conventional soil mapping. To see if BIS-4D can identify these, we selected properties and depth slices to represent the profile, these properties corresponding to expected diagnostic horizons or series differences in the test area."
- 11. Line 272: please expand on "method scales well".

- >> Reply: This is indeed an imprecise statement, and the section title is too broad we didn't discuss scaling as such. We propose to replace the section head with "Segmentation over a large area", and the topic sentence to begin with "To determine whether segmentation could be applied over a larger area than the 40~x~40~km test area, we segmented the BIS-4D product for the entire land area of the Netherlands..."
- 12. Figure 11: the black lines over dark background are very hard to see, please consider lighter colors or some other background color ramp. I think that main themes of this manuscript depend heavily on visual clarity within figures.
- >> Reply: Correct, this was a design fault. The figure will be redone with a lighter palette: "YlGnBu" from RColorBrewer, and the black border lines will then be a good choice. However, we also took the opportunity to reconsider this Figure and \S3.2.5 "Segmentation parameters". We feel a more effective figure is showing the effect of the easier segmentation (higher thresholds) at two shift sizes, in a 2 x 2 plot. The message of the \S will remain the same, i.e., the large effect of thresholds. Figure 10 (whole-county segmentation) will also be re-done, same colour scheme but orange segmentation boundaries.
- 13. Figure 12: There is a lot going on in this figure——I think that with some work it could be much simpler to interpret. See my hand—written notes in the attached PDF. In short, carefuly annotation and links back to the text are essential to making the figure relevant. It is currently very hard to understand the purpose.
- >> Reply: This and Figure 23 are the dendrograms. We intend to modify them according to the suggestions: annotate clusters with their numbers, remove meaningless coloured boxes, mark cut points mentioned in text. We will experiment with a horizontal layout as suggested (as in `aqp`) and compare the interpretability.
- 14. Figure 13: Are all segmentations the same? If so, it would help to make that clear in the caption. The spatial structure is very hard to "see" using the segment IDs, are there other ways to encode this information with color? For example, using greyscale for the background soil properties and semi-transparent color for the segment IDs...?
- >> Reply: Yes, they are the same. That can be seen by the maps themselves and the segement numbers. Colour is already used to show the values of the soil properties. Using colour for the IDs instead of the properties would result in an unworkable number of colours. Also they would not be connotative, the segment numbers are arbitrary from the algorithm. We think the text numbers can be read and referred back to the hierarchical clustering.

We propose to modify the caption to "Seven generalised clusters of the segmentation of Figure 9, based on slicing the clustering

- dendrogram shown in Figure 12. The same segmentation is shown for nine selected property-depth combinations."
- 15. Figure 14: it is not clear what subregion of this area was used in subsequent analysis and figures—please annotate with a rectangle. The color scale needs units and label.
- >> Reply: The units are given in the caption, but we will also add them above the color scale. We will outline the smaller area used for aggregation and point this out in the caption. It is exactly in the centre of the larger area.
- 16. Lines 345-350: could the apparent heterogenaity in SOC predictions be an artifact of imagery used to train the statistical model? In other words, is it logical to assume that the spectral signature of center-pivot irrigation would is monotonically associated with soil organic carbon? That might be true of some crops, but not all—especially leafy vegetables where most of the carbon would be in foliage vs. roots.
- >> Reply: We have not examined the SoilGrids predictions with interpretable machine learning methods (such as Shapley values for selected predicted locations) which might reveal this. That is outside the scope of this paper, which takes the DSM product as a given and evaluates it without attempting to relate the results to the methods used to make the product. Indeed it may be that the global model used in SoilGrids uses vegetation indices or equivalent from multivariate imagery as a key part of the SOC model.
- 17. Figure 18: it looks to me like there is no meaningful generalization of the landscape at this scale and with these segmenting parameters. If that is the author's interpretation then it should be more clear in the narrative. Maybe I missed something...?
- >> Reply: Lines 356-7 state "The level of detail is apparent, but many segments seem to be of a single class, with no internal pattern. Broader landscape patterns are obscured by this level of detail." Which seems to be what the reviewer is also noticing.
- 18. Line 360: please explain "useful at their respective design scales".
- >> Reply: Yes, this was an imprecise statement, and will be removed. The previous sentences already imply this.
- 19. Figures 19, 20, and 21: see my handwritten notes about legibility, units of measure, and color scales.
- >> Reply: We propose to adjust the figures as suggested. Figure 19: single legend, larger maps. Figure 20: as the suggestions for Figure 23 (see below).
- 20. Figure 22: same comments as Figure 12.

- >> Reply: Same reply.
- 21. Figure 23: similar comments as Figure 13—it is very difficult to compare segmentation using only the aggregate SOC values. Please consider coloring the segments with transparent colors, and using a greyscale color scale for the background values. Also, try to callout segment #1 (referenced in the narrative) using thicker outlines or special fill.
- >> Reply: Since the segmentation is based on SOC we feel that it is an appropriate background, and if just grey scale would be less apparent. The purpose here is just to show the J-S divergence with text. Yes, segment 1 should be highlighted and that explained in the caption. However as we revise the paper we intend to implement these suggestions and evaluate the revised figure.
- 22. Lines 383-388: does the segmentation follow any kind of recognizable pattern? If so or if not, please make the assertion stronger via example.
- >> Reply: Yes, most segments correspond to landscape units: Rio Grade valley, Mescalero moutains, West Texas/Eastern NM plateau, Chihuahuan mountains. A sentence will be added to highlight this success. Proposed text:
- "This was most apparent at the 1:16M nominal resolution (Figure 21), Among the most obvious are the Chihuahuan basin—and—range mountains (segment 29 of Figure 21), the upper Rio Grande valley near Soccorro (segment 2), and the west Texas/eastern New Mexico plateau (segment 13). Some segments include several physiographic units, which nonetheless apparently had similar patterns of SOC, for example segment 23 which includes some west Texas uplands, the Rio Grande valley below El Paso, and uplands in eastern Chihuahua."
- 23. Lines 395-400: I'd suggest a link to the maps vs. citation of the hosted data.
- >> Reply: The hosted data allows the reader to download the maps and open in their own GIS/R spatial.
- 24. Lines 400-405: consider adding the soil survey cartographic scale and order. See my hand-written notes here.
- >> Reply: In fact there is part of another county, so both will be referenced, with order and scale: "Wayne (Higgins, 1978) and Ontario (Pearson, 1958) Counties NY, originally published in 1978 and 1958, as Order 2, 1:15~840 and 1:20~000 scale surveys, respectively, on an unrectified airphoto base, and later digitised on a topographic base map by the NRCS and incorporated into gSSURGO. "Soil survey reports will be in the reference list.
- 25. Lines 408-412: "regression to the mean" is expected with all statistical models, why limit to "DSM products". It helps connect

our work to a wider range of applications and disciplines when we define term and outcomes with clear language.

- >> Reply: Since we include DSM products that don't use statistical learning (in the Hastie & Tishibiri sense), e.g., similarity and fuzzy matching methods, we can't make this statement about all DSM products. We do know that SOLUS is made by machine learning, so here we propose to modify the statement as "... SOLUS predicts a narrower range of concentrations. This is typical of DSM products, such as SOLUS, made with statistical learning methods (Hastie et al., 2009). This gives us a chance to cite that excellent text.
- 26. Lines 410-420: I suggest defining MLD and OLD early in the manuscript. These are incredibly important concepts that should be clear to the reader while they are reading through the manuscript.
- >> Reply: MLD was defined at line 154, "i.e., the smallest area that can be displayed on a printed map, of 0.25 cm2 at map scale, i.e., a grid cell side of 0.5 cm (Vink, 1963)." Indeed OLD was not defined until L415. We propose to move this definition to immediately after the L155 definition of MLD.
- We propose: "The Optimal Legible Delineation (OLD) is conventionally defined as 4 x MLD \citep{Forbes.etal1982}. This is a delineation size which is easily legible and still small enough to be relatively homogeneous. In conventional mapping the map scale should be set so that the soil pattern is on average able to be shown by OLD-sized polygons. In segmenting DSM products we hope that most segments are at least as large as the OLD."
- 27. Figure 24: these maps need to be larger and ideally us a less saturated color ramp, it is very hard to see differences < 20%. The subfigures should use a common color scale, and be in the same coordinate reference / grid system.
- >> Reply: We propose to make these changes as suggested.
- 28. Line 424: what do the SOLUS data appear to represent, spurious influence from e.g. elevation?
- >> Reply: Not elevation, that is a uniform range over this area (Lake Ontario plain), with local relief (drumlins and swamps) but no trend, and no obvious difference between the SE quadrant of SOLUS with the higher predicted clay, and the central N section with lower predicted clay. The tops of the drumlins are consistenly around 500' and the swamps around 380'. Land use (i.e., vegetation cover) is also similar throughout the area, depending on the local topography (drumlins and swamps). It's not clear what SOLUS might be representing. Looking at the Cropland Data Layer (https://www.nass.usda.gov/Research\_and\_Science/Cropland/Viewer/index.php) there seems to be some association with active cropland or apples (the higher predicted clay areas) and deciduous forests on the same landforms (the lower predicted clay areas), but this is only our visual assessment and may not be what SOLUS is picking up.

- We propose to add a sentence here: "After examining the supercells pattern and the source map, it is unclear to us what the SOLUS model is ``seeing'' in this area."
- 29. Figure 25: this map is very hard to interpret due to line density and overprinting of red/blue lines. Would it be possible to call-out 2 subregions of this area for a more detailed view into what is happening here?
- >> Reply: Yes, this is a good idea. We will compare a pure drumlin field with the central area with drumlins and inter-drumlin swamps.
- 30. Figure 26: figures are too small and should share a common color ramp. The caption should include more description of what the colors represent.
- >> Reply: Agreed, will make these changes.
- 31. Figure 27: figures are too small and should share common color ramp. What are the units of measure? The black segmentation lines are impossible to see on the right-most subfigure.
- >> Reply: The units of measure were mentioned in the text but will be added to caption. Segmentation lines will be drawn in dark orange for better contrast.
- 32. Line 444: I understand what the authors are saying here, but it would help to formalize, early on, how the reader should interpret "speak for itself". A more precise wording is proposed to replace this sentence: "The two algorithms applied to SOLUS100, with appropriate parameters, allowed the product ``speak for itself'', but the message was not clear and even misleading."
- >> Reply: Following another reviewer's suggestion, the title is modified to "helping the map..." to emphasize that the analyst has to make many choices when using the aggregation and segmentation algorithms.
- 33. Line 449: could this be another example of the "global model local interpretation" quagmire?
- >> Reply: Definitely. We propose to add a sentence making this point explicit, with a reference.
- "This is not meant to be a condemnation of SOLUS as a useful product overall. All DSM models trained over a wide area can have difficulty when applied to a local area with idiosyncratic soil—landscape relations which are not reflected in the covariates available over the entire training area, or which have locally—specific relations with the wider—area covariates. This is a general 'global model applied to locally—idiosyncratic landscapes' issue. This problem was already recognized early on in DSM exercises. For example Poggio et al. (2010) discovered that soil available water capacity models used

different significant covariates according to the level in a hierarchy of national (Scotland), regional and catchment, and recommended fitting models at the target extent. So in this study area, perhaps fitting the SOLUS model locally would have been more successful in reproducing the soil landscape pattern, even without local covariates related to glaciation."

- 34. Lines 470-480: could it be that SOC patterns in space are just too "fine" for this kind of evaluation? I think so.
- >> Reply: Indeed SOC patterns are fine right down to the sub-pedon level. However the support of most (?) observations that are used in SoilGrids was at least the pedon, and in practice most sampled pedons are located in "representative" sites. SoilGrids predicts at this support, at the centre of the grid cell ("pixel"). We (and DSM in general) suppose that the pedon-level observation is sufficiently representative of its pixel, and that if the sampling was done properly, material for the laboratory analysis was a composite across the face of the profile, so at least the 1 m scale variability has been removed.
- 35. Figure 28: how is this figure helpful to this final sections of the paper? Please explain and discuss. The map also needs a color scale with label and units of measure.
- >> Reply: This is explained in the paragaph (L471-479) where the map is referenced. It shows the reason why we can't discover a "true" SOC pattern with which to compare the SoilGrids SOC map. The colour bar from the interactive GLOSIS website will be added to the figure. The units are T ha $^{-1}$  SOC, this will be added to the caption.
- 36. Lines 495–500: do the authors have any parting words about \_how\_ we should proceed with evaluating the predictions of spatial models, beyond point-based methods? Several were demonstrated and hinted at within the manuscript, but it would be useful to expand on those here.
- >> Reply: We propose to add a final paragraph to \S6 "Discussion" explaining how the information from aggregation and segmentation might be used in an overall evaluation. First proposed sentence "So, how should aggregation and segmentation be used in an overall evaluation of a DSM product?"

**Paragraphs to follow:**

- "The common use of point evaluation statistics by cross-validation or repeated data splitting is still important, as long as the representativeness in both geographic and feature space is clear to the map user...."
- "An obvious evaluation of aggregation and segmentation can be the expert opinion of the soil geographer familiar with the mapped area..."
- "The starting point in any evaluation is the intended use(s) of the map..."
- "One application where segmentation analysis must be used is

identifying areas similar in their internal spatial pattern to a known area where the pattern has been characterized...."

**Comments made on the PDF (marginal notes, figure annotations)**

These can't be seen by the community. Most are also included above under "Detailed comments" and others are expansions of these. Some that go beyond this:

- 1.. L314 "Political boundaries are nowhere visible, except where one or more covariates match these" is marked as irrelevant.
- >> Reply: We think this is a major advantage of "top-down" modelling at global scale, so we propose to retain the statement.
- 2.. L188 Why is J-S divergence "stable"?
- >> Reply: This is explained in the cited Lin (1991) paper, the citation used above in the text can be added here. We propose to reword "stable" as "not sensitive to extreme values".